# Tail engagement of arrestin at the glucagon receptor

Kun Chen[1,2,8], Chenhui Zhang[1,2,8], Shuling Lin[1,8], Xinyu Yan[3], Heng Cai[4], Cuiying Yi[1], Limin Ma[1], Xiaojing Chu[1], Yuchen Liu[1,2], Ya Zhu[5], Shuo Han[1,2,4], Qiang Zhao[1,2,3,6]✉ & Beili Wu[1,2,4,7]✉

Arrestins have pivotal roles in regulating G protein-coupled receptor (GPCR) signalling by desensitizing G protein activation and mediating receptor internalization[1,2]. It has been proposed that the arrestin binds to the receptor in two different conformations, 'tail' and 'core', which were suggested to govern distinct processes of receptor signalling and trafficking[3,4]. However, little structural information is available for the tail engagement of the arrestins. Here we report two structures of the glucagon receptor (GCGR) bound to β-arrestin 1 (βarr1) in glucagon-bound and ligand-free states. These structures reveal a receptor tail-engaged binding mode of βarr1 with many unique features, to our knowledge, not previously observed. Helix VIII, instead of the receptor core, has a major role in accommodating βarr1 by forming extensive interactions with the central crest of βarr1. The tail-binding pose is further defined by a close proximity between the βarr1 C-edge and the receptor helical bundle, and stabilized by a phosphoinositide derivative that bridges βarr1 with helices I and VIII of GCGR. Lacking any contact with the arrestin, the receptor core is in an inactive state and loosely binds to glucagon. Further functional studies suggest that the tail conformation of GCGR–βarr governs βarr recruitment at the plasma membrane and endocytosis of GCGR, and provides a molecular basis for the receptor forming a super-complex simultaneously with G protein and βarr to promote sustained signalling within endosomes. These findings extend our knowledge about the arrestin-mediated modulation of GPCR functionalities.

In response to a vast array of agonists, GPCRs activate heterotrimeric G proteins to initiate various downstream signalling pathways[5]. To avoid overstimulation, a GPCR kinase (GRK) induces phosphorylation in the C-terminal region and/or intracellular loops of the receptor to trigger the recruitment of βarr, βarr1 or βarr2, which couples to the receptor in a similar binding site to the G protein-binding site and thus terminates G protein signalling[3,6,7]. Following the desensitization of G protein activation, βarr further promotes internalization of the GPCR–βarr complex[8]. On the basis of the trafficking itineraries after internalization, the GPCRs are categorized into two classes: 'class A' receptors internalize alone after a transient interaction with the arrestin and recycle rapidly to the plasma membrane, whereas 'class B' receptors intend to undergo sustained internalization into endosomes with the arrestin bound[9,10]. A previous negative-stain electron microscopy analysis of the complex between βarr1 and a C terminus-modified β2 adrenergic receptor (β2AR), β2V2R, revealed two distinct binding poses of βarr1, including a tail conformation that binds solely to the phosphorylated C-terminal tail of the receptor and a core conformation coupling to both the receptor transmembrane core and the C terminus[3]. Further evidence has suggested that these two binding patterns have differential roles in arrestin activation, cellular trafficking and subsequent

cellular responses[4,11,12]. These findings highlight the complexity of the arrestins in the modulation of GPCR function. However, the currently available arrestin-bound structures of GPCRs, which all belong to the rhodopsin-like GPCR family, adopt the core conformation[7,13–18]. Lack of molecular details of the tail conformation hampers our understanding of the arrestin-mediated regulation of GPCRs. More structural data of receptor–arrestin interaction, especially for other GPCR families, are essential to fully decipher the molecular mechanisms of GPCR signalling.

The secretin receptor family, including GCGR, exhibits many unique features in term of ligand recognition and receptor activation[19,20]. These receptors can activate multiple G proteins as well as the βarrs, resulting in distinct physiological processes[21]. Thus, they are of great interest as targets for developing biased agonists, which preferentially stimulate either the G protein-dependent pathways or βarr recruitment, as potential therapeutics for the treatment of type 2 diabetes and osteoporosis, among others[21–26]. Furthermore, it has been reported that some receptors in this GPCR family not only activate the G proteins at the plasma membrane but also promote sustained G protein signalling even after internalization into endosomes, leading to additional physiological consequences[27–31]. This adds complexity to the mechanisms of receptor

[1]State Key Laboratory of Drug Research, State Key Laboratory of Chemical Biology, Shanghai Institute of Materia Medica, Chinese Academy of Sciences, Shanghai, China. [2]University of Chinese Academy of Sciences, Beijing, China. [3]School of Chinese Materia Medica, Nanjing University of Chinese Medicine, Nanjing, China. [4]School of Pharmaceutical Science and Technology, Hangzhou Institute for Advanced Study, University of Chinese Academy of Sciences, Hangzhou, China. [5]Lingang Laboratory, Shanghai, China. [6]Zhongshan Institute for Drug Discovery, Shanghai Institute of Materia Medica, Chinese Academy of Sciences, Zhongshan, China. [7]School of Life Science and Technology, ShanghaiTech University, Shanghai, China. [8]These authors contributed equally: Kun Chen, Chenhui Zhang, Shuling Lin. ✉e-mail: zhaoq@simm.ac.cn; beiliwu@simm.ac.cn

signalling and raises the possibility of developing biased ligands that specifically target different phases of signalling.

To uncover molecular details of βarr in modulating receptor signalling and facilitate biased ligand discovery for the secretin receptor family, we determined the structures of GCGR–βarr1 complex and performed extensive functional studies. This work provides a detailed picture of the interaction pattern between the GPCR and the arrestin in a tail conformation, and discloses key factors that govern the cellular trafficking and sustained signalling of GCGR.

## Structure determination of GCGR–βarr1

To facilitate complex formation, the C-terminal region of GCGR (residues H433–F477) was exchanged for the C-terminal residues A343–S371 of the vasopressin type 2 receptor ($V_2R$) (termed GCGR($V_2$RC)). Aiming to improve complex stability, a cysteine-free βarr1 was generated by introducing seven mutations and its C-terminal region (residues I377–R418) was replaced with the antibody scFv30. Supported by our functional study using a bioluminescence resonance energy transfer (BRET) assay, which measures the proximity between the C termini of GCGR and βarr1, these protein modifications have little effect on receptor–arrestin interaction (Extended Data Table 1). To obtain the intact GCGR($V_2$RC)–βarr1 complex, the receptor and βarr1 were co-expressed together with GRK2 and co-purified in the presence of the endogenous agonist glucagon. The protein sample was then subjected to cryo-electron microscopy (cryo-EM) single-particle analysis, yielding two maps of the complex in glucagon-bound and ligand-free states at resolutions of 3.3 Å and 3.5 Å, respectively (Fig. 1a,b, Extended Data Table 2 and Extended Data Fig. 1). The maps allowed unambiguous modelling of the majority of the residues in the receptor transmembrane domain and βarr1 (Extended Data Fig. 2). Except for the distinct ligand-binding states, the two structures are similar with a root-mean-squared deviation (r.m.s.d.) of 1.4 Å for all atoms.

## The tail engagement of βarr1 at GCGR

Despite differences in protein modification (C-terminal tail, truncation and mutation, among others) and sample preparation (detergent, nanodiscs and antibody, among others), the previously published structures of the GPCR–arrestin complexes all exhibit a core conformation of the arrestin with its finger loop penetrating into the intracellular pocket of the receptor helical bundle[7,13–18]. In our GCGR($V_2$RC)–βarr1 structures, the receptor interacts with βarr1 mainly through its C-terminal region, including helix VIII and the $V_2R$ tail, whereas the receptor intracellular pocket remains unoccupied (Fig. 1c,d). This observation indicates that βarr1 is in a tail conformational state. However, unlike the tail conformation previously observed in the negative-stain EM study of the $\beta_2V_2R$–βarr1 complex, where βarr1 appears to hang from the receptor with its long axis perpendicular to the membrane plane[3], the arrestin forms an approximately 45° angle with the membrane plane upon binding to GCGR($V_2$RC) (Fig. 2a). This allows βarr1 to make notably more contacts with the receptor and membrane in multiple regions. Instead of binding to the receptor core, the loops in the central crest of βarr1, including the finger loop, form extensive interactions with helix VIII of GCGR (Fig. 2b). The phosphorylated $V_2R$ tail binds to the N-lobe groove of βarr1 through charge complementarity interactions as previously observed[14,16,17] (Fig. 2c). The C-edge of βarr1, which does not make any direct contact with the receptors in the previous GPCR–arrestin structures, is adjacent to the first intracellular loop (ICL1) and the intracellular tip of helix IV of GCGR, and stabilizes the tail-binding pose by being embedded in the membrane layer (Figs. 1a–d and 2d). The tail engagement of the GCGR($V_2$RC)–βarr1 complex unlikely results from the C-terminal $V_2R$-tail replacement of GCGR, as all the previously determined arrestin-bound GPCR structures adopt the core conformation despite different C-terminal tails in those receptors (with or without the $V_2R$ tail). The differences between the observed tail and core conformations imply diversity of the arrestin binding modes in recognition of different GPCRs, which may be family specific.

Compared with the core conformation in the previous GPCR–arrestin structures, the arrestin in the GCGR($V_2$RC)–βarr1 complex is parallel to those in the arrestin-bound structures of rhodopsin, the $\beta_1$ adrenergic receptor ($\beta_1$AR) and the M2 muscarinic receptor (M2R), but forms an angle of 20–50° between the long axes with those in the βarr1 complexes of neurotensin receptor 1 (NTSR1), $V_2R$ and the 5-HT$_{2B}$ serotonin receptor (Fig. 1e,f and Extended Data Fig. 3a,b). Upon binding to GCGR($V_2$RC), the centre of βarr1 shifts along helix VIII by 37–46 Å (measured at the Cα atom of D135 in the middle loop of βarr1 (D139 in the visual arrestin)) due to the fact that the central loops of βarr1 interact with helix VIII instead of the helical core, which reflects the major difference between the tail and the core conformations (Fig. 1e,f and Extended Data Fig. 3a,b). The βarr1 in the GCGR($V_2$RC)–βarr1 structures is structurally similar to the arrestins in the other known GPCR–arrestin structures, with a Cα r.m.s.d. of 1.3–1.7 Å (GCGR($V_2$RC)–βarr1 versus others), indicating that the receptor tail-engaged arrestin is also in the active state. The largest deviation occurs in the central loops and C-edge, which aligns with different interaction patterns in these regions when bound to different receptors (Extended Data Fig. 3c).

The βarr1-bound GCGR structure was also compared with our previously determined structures of the glucagon–GCGR–G$_s$ and glucagon–GCGR–G$_i$ complexes[20]. Superposition of the receptors in these structures reveals a major overlap of βarr1 with the Gβγ subunits, but only a partial overlap between the βarr1 C-edge and the αN helix in the Gα subunit (Fig. 1g). The largely distinct binding sites of βarr1 and Gα at GCGR may provide a molecular basis for the formation of a super-complex of the receptor bound simultaneously to both the G protein and the arrestin (discussed below).

## Interactions between GCGR and βarr1

The tail conformation of arrestin was believed to be solely mediated by the phosphorylated C-terminal tail of the receptor[3,4]. Unexpectedly, in the GCGR($V_2$RC)–βarr1 complexes, helix VIII of the receptor has a major role in defining the tail-binding pose of βarr1 (Fig. 2a). The segment of H416–W425 in the C-terminal region of helix VIII interacts with the central loops of βarr1 mainly through hydrophobic contacts, with the residues L420, V423, L424 and W425 forming a hydrophobic patch and fitting into a shallow groove shaped by the finger loop (residues 63–75), middle loop (residues 129–140), C-loop (residues 241–249) and lariat loop (residues 274–300) in βarr1 (Fig. 2b). The interaction in this region is mediated by two hydrophobic cores, including one established by the receptor residues L420, V423 and L424 and the βarr1 residues Y63, L129, I241, L243, A247 and Y249 in the finger loop, middle loop and C-loop, and the other one formed between the bulky residue W425 in helix VIII and the βarr1 residues L129, Y249, R285 and G286 in the middle loop, C-loop and lariat loop (Fig. 2b). In addition, two hydrogen bonds between the side chains of the GCGR residue H416 and the residue N245 in the C-loop of βarr1 as well as between the side chain of R417 in the receptor and the main chain carbonyl of Q248 in βarr1 are also observed, further strengthening the helix VIII–central crest binding (Fig. 2b). The importance of these interactions in arrestin binding was supported by mutagenesis studies using the BRET assay, showing that the alanine or tryptophan mutations of most of the key residues reduced glucagon potency (the half-maximal effective concentration (EC$_{50}$)) in triggering βarr1 coupling by over tenfold (Fig. 2f, Extended Data Table 1 and Extended Data Fig. 4a–d). Among the mutations, the alanine substitutions of W425 (GCGR) and R285 (βarr1) that form a π–cation interaction display the largest effect by almost abolishing the binding (Fig. 2b,f and Extended Data Fig. 4a,d).

The requirement of helix VIII of GCGR for arrestin binding is consistent with a previous study that showed that deletion of helix VIII and

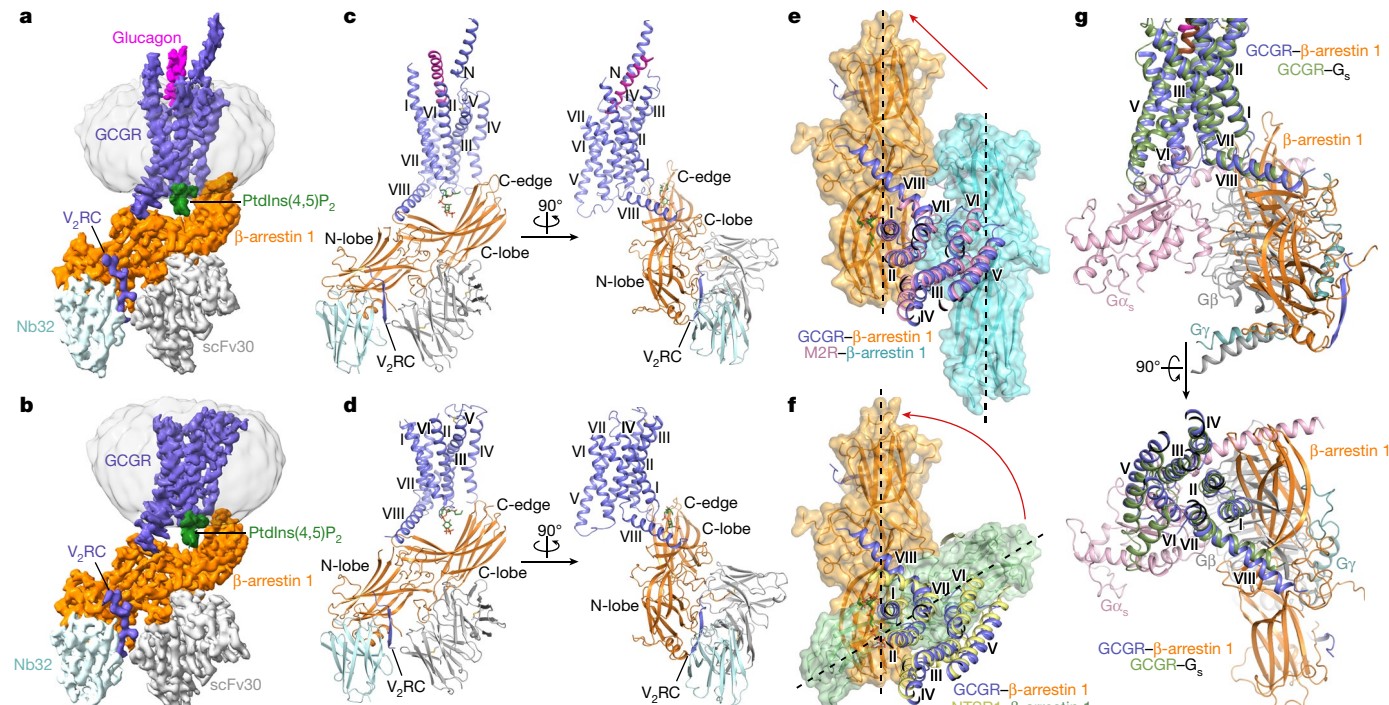

**Fig. 1 | Overall structures of the GCGR–βarr1 complexes and structural comparison. a,b,** Cryo-EM maps of the glucagon–GCGR(V₂RC)–βarr1 (**a**) and GCGR(V₂RC)–βarr1 (**b**) complexes coloured according to chains. **c,d,** Structures of the glucagon–GCGR(V₂RC)–βarr1 (**c**) and GCGR(V₂RC)–βarr1 (**d**) complexes. The structures are shown in two orientations. **e,f,** Comparison of the βarr1 binding modes in the glucagon–GCGR(V₂RC)–βarr1 structure and previously determined GPCR–βarr1 structures. Comparisons are shown between the glucagon–GCGR(V₂RC)–βarr1 and M2R–βarr1 (PDB ID: 6U1N) structures (**e**), and between the glucagon–GCGR(V₂RC)–βarr1 and NTSR1–βarr1 (PDB ID: 6UP7) structures (**f**). The structures are shown in an extracellular view, with the transmembrane domains aligned. The βarr1 in the structures is also shown in surface representation. The long axes of βarr1 are indicated by black dashed lines. The red arrows indicate the conformational movement of βarr1 in the glucagon–GCGR(V₂RC)–βarr1 structure relative to the previously determined GPCR–βarr1 structures. **g,** Comparison between the glucagon–GCGR(V₂RC)–βarr1 and glucagon–GCGR–Gₛ (PDB ID: 6LMK) structures. The structures are shown in membrane (top) and extracellular (bottom) views, with the transmembrane domains aligned.

the C terminus of the rat GCGR abolished receptor internalization, but removal of only the C terminus did not[32]. Using crosslinking and molecular dynamics simulations, a recent study has suggested that helix VIII of the secretin-like parathyroid hormone 1 receptor (PTH1R) participates in βarr1 binding; despite that, a core conformation of the arrestin was proposed[33]. The involvement of helix VIII in transducer binding was also observed in the previously determined G protein-bound structures of all the receptors in the secretin receptor family, but not in the G protein complexes of the other GPCRs. Together with these data, our GCGR(V₂RC)–βarr1 structures suggest a unique role of helix VIII in transducer recognition for this GPCR family.

In contrast to the previously reported GPCR–arrestin structures where the arrestin finger loop has a central role in mediating receptor recognition by forming interactions with the transmembrane core, upon binding to GCGR, the finger loop of βarr1 in the tail conformation only forms an interaction with the receptor through the residue Y63 in its N-terminal region (Fig. 2b). Lacking contacts with the receptor, the turn of the finger loop (residues 66–73) adopts a flexible conformation and was not traced in the structures. However, if the entire turn region (residues 64–77) was removed, a 24-fold reduction of EC₅₀ was observed in the BRET assay (Fig. 2f, Extended Data Table 1 and Extended Data Fig. 4b). This may result from a disturbance of the conformation of the βarr central crest and/or an impairment of another possible arrestin-binding pattern, such as a core conformation.

Interactions between the arrestin C-edge loops and detergent micelles or nanodiscs were observed in all the previously determined GPCR–arrestin structures. It has been suggested that the C-edge is critical for stabilizing the core conformation of receptor–arrestin complexes and may increase arrestin concentration at the cell membrane

to facilitate desensitization of G protein activation by anchoring to the plasma membrane[7,14,15]. The C-edge membrane interaction also exists in the GCGR(V₂RC)–βarr1 structures. However, in contrast to the other arrestin-bound structures, in which the arrestin C-edge is far away from the receptor, the GCGR(V₂RC)–βarr1 complexes display a close proximity of the βarr1 C-edge to the receptor helical bundle, with one of the loops (residues 189–195) forming contacts with ICL1 and the intracellular tip of helix IV in GCGR (Fig. 2d and Extended Data Fig. 3d). These extra interactions stabilize the positioning of the C-edge and help to define the tail-binding pose of the arrestin and subsequent cellular responses. This finding further highlights the importance of the C-edge–membrane anchoring in governing arrestin functionality.

Previous structural and functional studies have suggested that the membrane phosphoinositides are involved in modulating GPCR function by stabilizing the receptor–arrestin complexes[15,17,34]. This is further supported by our GCGR(V₂RC)–βarr1 structures, which exhibit a different binding mode of the phospholipid from that previously observed in the NTSR1–βarr1 structure[15] (Extended Data Fig. 3e). The cryo-EM maps display the densities for the phospholipid dioctyl-phosphatidylinsitol-4,5-bisphosphate (diC8-PtdIns(4,5)P₂), which was added during protein purification (Extended Data Fig. 2). It bridges the C-lobe of βarr1 with the intracellular tip of helix I, ICL1 and helix VIII of GCGR, acting as a 'trestle' to further stabilize the tail conformation of the complex (Fig. 2e). Similar to what was observed in the NTSR1–βarr1 structure, the 4,5-bisphosphate group of diC8-PtdIns(4,5)P₂ forms ionic interactions with multiple basic residues in the C-lobe of βarr1, including K232, R236, K250, K324 and K326, which have been reported as a binding site for inositol phosphates[35,36] (Fig. 2e). The requirement of the phospholipid for GCGR coupling to the arrestin was verified by the BRET

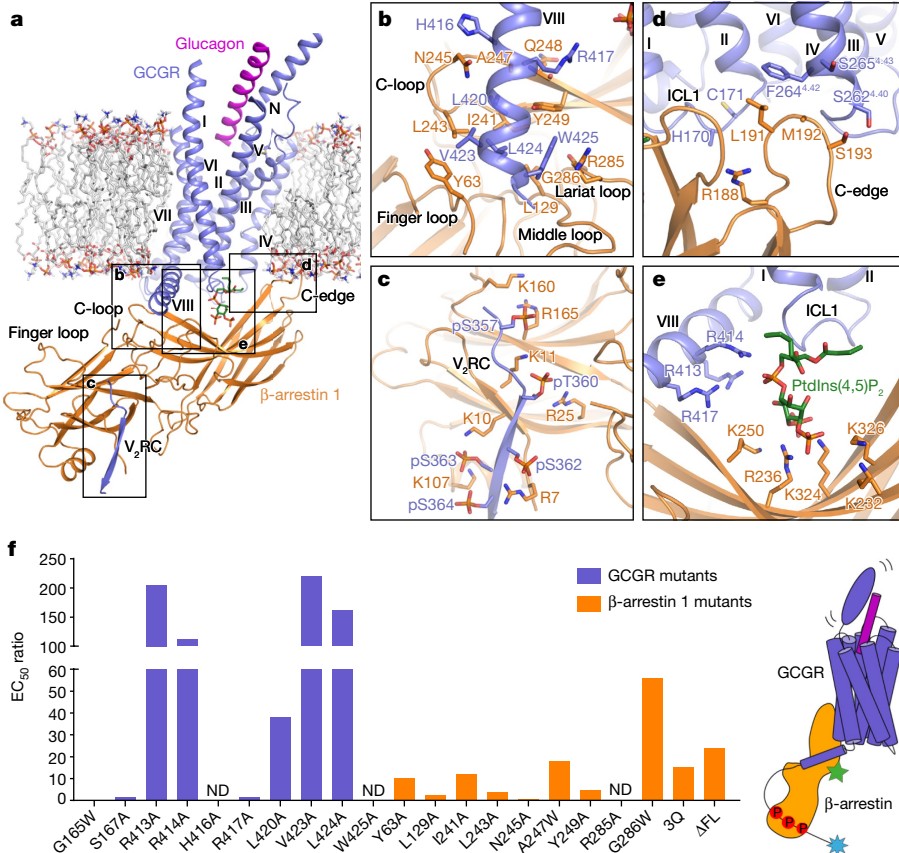

**Fig. 2 | Interactions between GCGR(V₂RC) and βarr1. a**, Overall view of the GCGR(V₂RC)–βarr1 interface. The main interaction sites are indicated by black boxes. The modelled lipid bilayer is shown as grey sticks. **b**–**e**, Enlarged views of the interactions in the main interaction sites. The key residues involved in the interactions are shown as sticks and coloured blue (GCGR) and orange (βarr1). Shown are: the interactions between helix VIII of GCGR and the central loops in βarr1 (**b**); the interactions between the V₂R tail (V₂RC) and the N-lobe of βarr1 (**c**); potential interactions between GCGR and the C-edge of βarr1 (**d**), in which the side chains of H170 (GCGR) and M192 (βarr1) are not modelled due to lack of electron densities; and the interactions mediated by the phospholipid

diC8-PtdIns(4,5)P₂ (**e**). **f**, Glucagon-induced GCGR–βarr1 interaction for the GCGR and βarr1 mutants measured by BRET assays. ΔFL, the βarr1 mutant with the turn region of the finger loop (residues 66–73) removed. The EC₅₀ ratios, EC₅₀(mutant)/EC₅₀(wild type), are represented by bars. Extended Data Table 1 provides detailed independent experiment numbers (*n*), statistical evaluation, *P* values and expression levels. A schematic of the GCGR–βarr complex in the tail conformation is also shown (right). The biosensors labelled in GCGR and βarr that were used in the BRET assay are indicated by blue and green stars, respectively. ND, not determined.

assay. The data showed that the combination of the mutations K232Q, R236Q and K250Q (3Q) in βarr1 resulted in a 15-fold drop of glucagon potency compared with that for the wild-type βarr1 (Fig. 2f, Extended Data Table 1 and Extended Data Fig. 4e).

In contrast to the similar binding mode between βarr1 and the phospholipid head group when bound to GCGR and NTSR1, the tail region of diC8-PtdIns(4,5)P₂ adopts distinct interaction patterns with these two receptors. In NTSR1, the membrane surface of helices I and IV is in close proximity to the phospholipid tail[15]. Upon binding to GCGR, the bridging phosphate of the phospholipid potentially interacts with three positively charged residues, R413, R414 and R417, in helix VIII of the receptor (Fig. 2e). The BRET data showed that the mutations R413A and R414A substantially impaired the glucagon-induced βarr1 binding with a 112–205-fold reduction of EC₅₀ and an about 50% drop of maximal response (*E*ₘₐₓ) (Fig. 2f, Extended Data Table 1 and Extended Data Fig. 4e), suggesting that these two residues have an important role in phospholipid binding. These two basic residues are conserved in the secretin receptor family, especially R413, which is arginine or lysine in all the receptors (Extended Data Fig. 5). This implies that a similar helix VIII–phospholipid interaction pattern may also exist in the other receptors of this GPCR family. In addition to helix VIII, the intracellular tip of helix I and ICL1 in GCGR also make close contacts with the tail of diC8-PtdIns(4,5)P₂ (Fig. 2e). However,

the mutations G165W and S167A in this region had little effect on βarr1 coupling (Fig. 2f and Extended Data Table 1). This may be explained by the dynamic nature of the phospholipid tail, which is reflected by weaker densities of this region than the head group in the cryo-EM maps (Extended Data Fig. 2).

## Inactive state of the βarr1-bound GCGR

Another difference in the GCGR(V₂RC)–βarr1 structures compared with the other known arrestin-bound structures is that GCGR adopts an inactive conformation even in the presence of the agonist glucagon, whereas the other receptors are in an active state. The transmembrane helical bundle of the βarr1-bound GCGR is structurally more similar to that in our previously determined inactive structure of GCGR bound to the inhibitor NNC0640 and the inhibitory antibody mAb1 (ref. 37) (Cα r.m.s.d. of 1.3 Å) than to the fully active structure of glucagon–GCGR–Gₛ[20] (Cα r.m.s.d. of 2.1 Å) (Fig. 3a,b). In the GCGR(V₂RC)–βarr1 structures, the intracellular region of helix VI, which undergoes a large outward movement in the G protein-bound GCGR structures, adopts a similar conformation to that in the inactive structure. This conformational feature of GCGR is most likely attributed to the tail-binding mode of the arrestin. Lacking any contact with the receptor core, the tail-engaged βarr1 does not require the receptor to retain its active conformation.

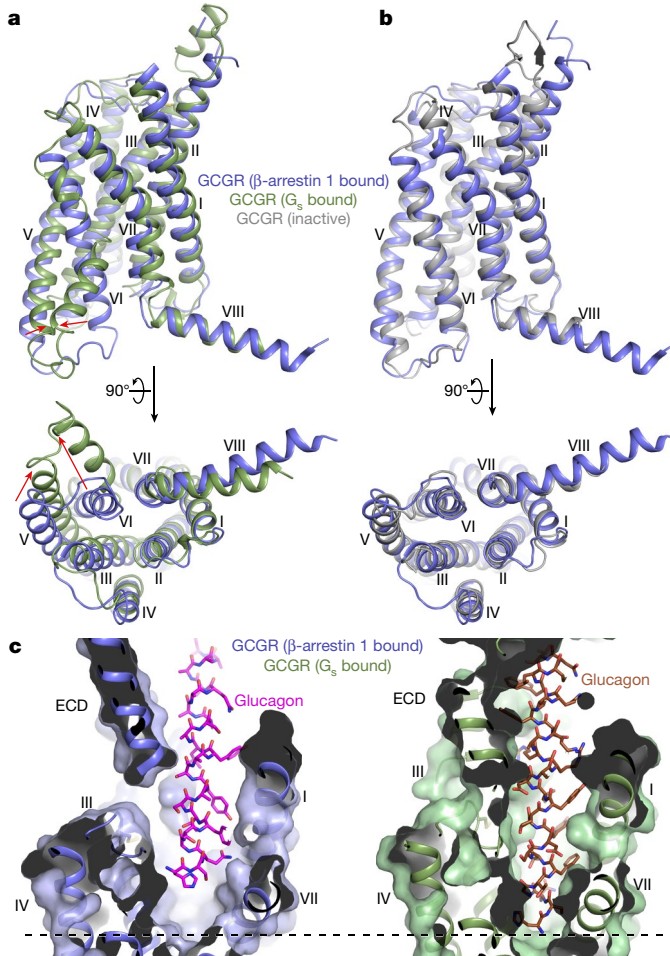

**Fig. 3 | Inactive conformation of the βarr1-bound GCGR. a**, Comparison of the GCGR transmembrane helical bundles in the glucagon–GCGR(V₂RC)–βarr1 and glucagon–GCGR–Gₛ structures. The red arrows indicate the conformational differences of helices V and VI in the Gₛ-bound GCGR relative to the βarr1-bound GCGR. **b**, Comparison of the GCGR transmembrane helical bundles in the glucagon–GCGR(V₂RC)–βarr1 and inactive NNC0640–GCGR–mAb1 (PDB ID: 5XEZ) structures. In **a**,**b**, the helical bundles in the structures are shown in membrane (top) and intracellular (bottom) views. **c**, Comparison of the glucagon binding modes in the glucagon–GCGR(V₂RC)–βarr1 and glucagon–GCGR–Gₛ structures. The receptors are shown in cartoon and surface representations. Glucagon is shown as sticks in both structures. The black dashed line indicates the bottom of the ligand-binding pocket. ECD, extracellular domain.

Consistent with the inactive conformation of the receptor, the agonist glucagon is either absent or loosely attached to the receptor in the βarr1-bound GCGR structures. Comparison with the glucagon–GCGR–Gₛ structure reveals a shift of the peptide towards the extracellular surface in the βarr1-bound complex (Fig. 3c). The different binding modes of glucagon in the two structures are associated with distinct rotamer conformations of the receptor residue R308[5.40] (the superscript refers to the Wootten numbering system[38]). In the glucagon–GCGR–Gₛ complex, the positively charged residue H1 of glucagon binds deep to the ligand-binding pocket and repels the side chain of R308[5.40] away from the ligand-binding pocket, whereas in the glucagon–GCGR(V₂RC)–βarr1 complex, the shift of the peptide makes space for the residue R308[5.40], allowing its side chain to point towards the centre of the helical bundle (Extended Data Fig. 3f).

The upward movement of glucagon breaks the receptor–peptide interaction network and thus impairs the stability of the GCGR–glucagon complex. This is supported by poor densities for the receptor

extracellular domain and the peptide C terminus and 3D variability analysis of the cryo-EM data, showing a larger motion of this region in the βarr1-bound complex relative to the Gₛ-bound complex (Extended Data Fig. 1a and Supplementary Videos 1 and 2). The loose receptor–peptide binding is probably associated with the empty intracellular pocket, given that the intracellular transducer protein coupling to the receptor core would provide an allosteric effect on stabilizing the agonist binding on the extracellular side[39]. Owing to the lack of interaction in the deeper region, the agonist is unable to trigger the conformational rearrangement of the helical bundle and subsequent receptor activation. However, in our functional studies, the agonist is required for maximal arrestin binding. This may be explained by the requirement of the agonist for triggering GRK binding for phosphorylation and/or stabilizing the receptor active conformation that is essential for other possible arrestin binding modes. The loosely bound glucagon in the tail-engaged GCGR–arrestin complex may further adopt the tight binding mode to the receptor to facilitate G protein activation after internalization into endosomes (discussed below).

## Tail conformation mediates trafficking

To study the cellular trafficking pattern of GCGR, we monitored recruitment of βarr to the plasma membrane and early endosome using the BRET biosensors *Renilla reniformis* green fluorescent protein (rGFP)–CAAX and GFP2–FYVE, respectively[4,40] (Fig. 4a,b). Glucagon induced an increase of the BRET signal between Rluc8–βarr2 and rGFP–CAAX as well as between Rluc8–βarr2 and GFP2–FYVE in HEK293F cells expressing the wild-type GCGR (Fig. 4c,d and Extended Data Table 3). This data indicate that GCGR is able to promote both βarr recruitment to the plasma membrane and sustained internalization into endosomes, and thus, GCGR qualifies as a class B receptor. A wild-type level of βarr membrane recruitment and endosome internalization was also observed for the chimeric GCGR(V₂RC), demonstrating that the C-terminal V₂R tail does not alter the cellular trafficking pattern of GCGR (Extended Data Table 3). To verify the reliability of the assays, we also measured the βarr recruitment at the plasma membrane and endocytosis for the angiotensin II receptor AT₁R, which has been classified as a class B GPCR[41,42], and the known class A receptor β₂AR[4,42]. As expected, AT₁R displayed an increase of the BRET signal in both assays, whereas β₂AR only exhibited an agonist-stimulated increase of the BRET signal between Rluc8–βarr2 and rGFP–CAAX, but not between Rluc8–βarr2 and GFP2–FYVE (Fig. 4c,d and Extended Data Table 3). It has been reported that some other members of the secretin receptor family are also able to promote sustained internalization into endosomes[29–31,43,44]. These data suggest that this GPCR family may have a common cellular trafficking feature.

Previous studies of β₂AR, β₂V₂R and V₂R have suggested that a βarr in the tail conformation is fully capable of promoting receptor internalization and signalling, whereas desensitization of G protein activation is exclusively mediated by the receptor core-engaged βarr[4,45]. To investigate the role of the tail-engaged GCGR–βarr in cellular trafficking, we performed mutagenesis studies using the BRET assays of plasma membrane recruitment and endocytosis. The alanine replacements of the key residues in helix VIII of GCGR that mediate the tail engagement of βarr, including H416, L420, V423, L424 and W425, reduced the maximal BRET signal between Rluc8–βarr2 and rGFP–CAAX by over 60% and decreased the glucagon potency in triggering endocytosis by over eightfold (except for L424A), with some of the mutations abolishing the signals (Fig. 4e,f and Extended Data Table 3). Furthermore, the GCGR mutation R413A and the 3Q mutation of βarr2, which disrupt the phospholipid binding to destabilize the tail-binding pose of βarr, also showed a drastic effect on both recruitment at the plasma membrane and endocytosis (Fig. 4e,f and Extended Data Table 3). These data strongly imply that the tail conformation of βarr is largely involved in the cellular trafficking of GCGR (Fig. 4i).

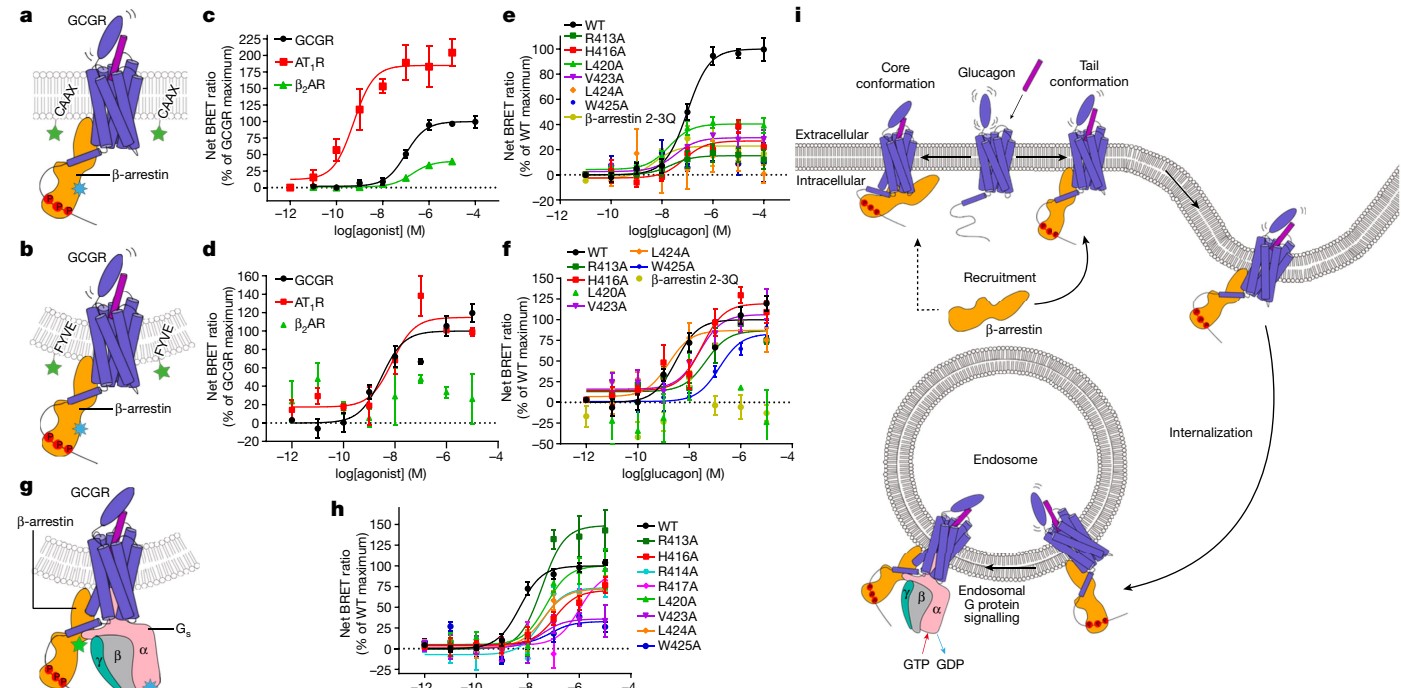

**Fig. 4 | The tail conformation of GCGR–βarr mediates cellular trafficking and megaplex formation. a,c,e**, Agonist-induced plasma membrane βarr recruitment measured by the BRET assay. Schematic of the tail conformation of the GCGR–βarr complex in the plasma membrane (**a**). The biosensors labelled in βarr and CAAX that were used in the BRET assay are indicated by blue and green stars, respectively. The plasma membrane βarr recruitment of the wild-type (WT) GCGR, AT$_1$R and β$_2$AR (**c**). The plasma membrane βarr recruitment of the WT GCGR and mutants (**e**). **b,d,f**, Agonist-induced endocytosis measured by the BRET assay. Schematic of the tail conformation of the GCGR–βarr complex within the endosome (**b**). The biosensors labelled in βarr and FYVE that were used in the BRET assay are indicated by blue and green stars, respectively. Endocytosis of the WT GCGR, AT$_1$R, and β$_2$AR (**d**). Endocytosis of the WT GCGR and mutants (**f**). The horizontal dotted lines in **a**–**f** indicate the base lines with the net BRET ratio as zero. **g,h**, Glucagon-induced G$_s$–βarr interaction measured by the BRET assay. Schematic of the G$_s$–GCGR–βarr megaplex (**g**). The biosensors labelled in Gα$_s$ and βarr that were used in the BRET assay are indicated by blue and green stars, respectively. The G$_s$–βarr interaction promoted by the WT GCGR and mutants (**h**). The data of the plasma membrane recruitment, endocytosis and the G$_s$–βarr interaction are shown as mean ± s.e.m. from at least three independent experiments performed in technical duplicate. Extended Data Tables 3 and 4 provide detailed numbers of independent experiments (*n*), statistical evaluation, *P* values and expression levels. **i**, Schematic representation of the functional processes mediated by the GCGR–βarr complex. The tail conformation has major roles in the plasma membrane βarr recruitment, internalization and megaplex formation. The core conformation may contribute to the plasma membrane recruitment to a lesser extent than the tail conformation.

Numerous GPCRs, including several receptors of the secretin receptor family, have been found to produce second messenger molecules in a sustained manner even after internalization into endosomes, and the majority of these receptors are class B GPCRs[27–29,31,46,47]. It has been proposed that the formation of a GPCR 'megaplex', in which the receptor binds to the heterotrimeric G protein with its transmembrane core and simultaneously couples to βarr through its phosphorylated C-terminal tail, provides a molecular basis for the sustained G protein signalling within endosomes[48–50]. To assess the ability of forming the megaplex, we utilized a BRET assay to measure the close molecular proximity between G$_s$ and βarr1 (Fig. 4g). Upon agonist stimulation, an increase in the BRET signal was observed for the wild-type GCGR, but not for β$_2$AR (Fig. 4h and Extended Data Table 4). The mutations in helix VIII of GCGR were further tested, showing that all the mutations substantially impaired the interaction between G$_s$ and βarr1, whereas these mutations had little effect on G$_s$ activation (Fig. 4h and Extended Data Table 4). These data suggest the existence of the G$_s$–GCGR–βarr megaplex and the importance of the tail engagement of βarr in mediating the formation of the megaplex (Fig. 4i). However, as mentioned previously, alignment of the βarr1-bound and G$_s$-bound GCGR structures reveals an overlap between βarr1 and the Gβγ subunits (Fig. 1g). This suggests that the binding patterns between GCGR and the transducers in the megaplex may be different from those in the complexes of the receptor bound to either of the transducers alone.

It was believed that the core conformation of βarr is responsible for desensitization of G protein signalling, as a spatial hindrance is required for an efficient blockade of G protein coupling[4,45]. We wondered whether such a conformation exists for the GCGR–βarr complex. Thus, on the basis of the receptor–arrestin interaction patterns in the previously reported GPCR–arrestin structures, we designed 20 single mutations in the intracellular surface of the helical bundle in GCGR, including in ICL2 and ICL3, the intracellular regions of helices II, III, V and VI, and the helix VII–VIII joint (Extended Data Fig. 4f,g). The receptor–βarr interaction was then measured for these mutants. Most of these mutations had little effect on βarr coupling, except for the mutations L329[5.61]A, K332[5.64]A, R336[ICL3]A and R346[6.37]A in ICL3 and helices V and VI (Extended Data Table 1 and Extended Data Fig. 4h), suggesting that this region may also be involved in βarr recognition and implying the potential existence of a core conformation. However, these mutations displayed much weaker effects on βarr recruitment at the plasma membrane and endocytosis (Extended Data Table 3 and Extended Data Fig. 4i,j), suggesting that the core-engaged βarr does not have a major role in the cellular trafficking of GCGR (Fig. 4i).

Together, this work provides molecular details of the βarr coupling to a GPCR in a tail conformation. Compared with the core conformation observed in the previous structural studies of the GPCR–arrestin complexes, the tail-engaged GCGR–βarr1 complex exhibits distinct features in the interaction pattern, phospholipid recognition, receptor

conformation and ligand binding, which provide a molecular basis for the arrestin defining the cellular trafficking and sustained signalling within endosomes. These findings underline the complexity of the mechanisms of the arrestin in governing receptor functionality and offer an opportunity for developing novel biased ligands with pathway selectivity.

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

# Methods

## Construct cloning and protein expression

To facilitate protein expression, the human *GCGR* gene was cloned into pFastBac1 expression vector with the endogenous signal peptide replaced with a haemagglutinin (HA) signal peptide, which was followed by a 12-residue epitope for the $Ca^{2+}$-dependent monoclonal antibody HPC4. The C-terminal residues H433–F477 of GCGR were replaced with the residues A343–S371 in the C terminus of $V_2R$[49]. To improve expression level and protein stability, a cysteine-free bovine βarr1 was generated by introducing the mutations C59A, C125S, C140I, C150V, C242V, C251V and C269S as previously described[14,15]. The preactivated mutation R169E was also introduced to increase the activation level of βarr1 (refs. 18,51). Furthermore, the C-terminal region of βarr1 (residues I377–R418) was replaced with an engineered single-chain Fab30 (scFv30) to stabilize the GCGR–βarr1 complex, and a 6× His tag was added to the C terminus.

The modified GCGR($V_2RC$) and βarr1 were co-expressed with GRK2 in *Spodoptera frugiperda* (*Sf*9) insect cells (Invitrogen; cells were routinely tested for mycoplasma contamination) using the Bac-to-Bac Baculovirus Expression System (Invitrogen). The cells were grown to a density of $1.5 \times 10^6$ cells per ml and infected with viral stocks of GCGR, βarr1 and GRK2 at a multiplicity of infection ratio of 6:2:2. The cells were cultured at 27 °C for 48 h and then harvested by centrifugation at 2,000$g$ for 20 min. The biomass was stored at −80 °C until use.

## Expression and purification of Nb32

Nb32 was prepared as previously described[4,52]. In brief, the gene encoding Nb32 was cloned into a pET28a vector with a PreScission protease site (LEVLFQGP) and an 8× His tag at the C terminus, and expressed in the *Escherichia coli* stain BL21(DE3). The cells were grown in LB medium supplemented with 50 μg ml$^{-1}$ kanamycin at 37 °C for 4 h and then cultured at 16 °C for 16 h after addition of 1 mM IPTG. The cells were then harvested by centrifugation at 3,000$g$ for 30 min and lysed in 20 mM HEPES (pH 7.5), 125 mM NaCl, 5 mM $MgCl_2$ and 2 mM DTT by sonication. The supernatant was collected by ultracentrifugation at 100,000$g$ for 30 min and loaded to Ni affinity chromatography (Clontech). The protein bound to the Ni resin was washed by a buffer containing 20 mM HEPES (pH 7.5), 125 mM NaCl, 5 mM $MgCl_2$ and 30 mM imidazole, and then cleaved by PreScission protease (custom made) at 4 °C for 18 h. The protein sample was collected and further purified on a Superdex 200 Increase 10/300 column (GE Healthcare) equilibrated with 20 mM HEPES (pH 7.5), 125 mM NaCl and 5 mM $MgCl_2$. The peak fractions of the protein sample were concentrated to about 10 mg ml$^{-1}$, flash-frozen in liquid nitrogen and store at −80 °C until use.

## Purification of the glucagon–GCGR($V_2RC$)–βarr1 complex

The cells expressing the GCGR($V_2RC$)–βarr1 complex were thawed on ice and suspended in phosphorylation buffer containing 20 mM HEPES (pH 7.5), 125 mM NaCl, 5 mM $MgCl_2$, 10 μM glucagon, 1 mM ATP, 2 mM $Na_3VO_4$ and EDTA-free protease cocktail inhibitor (Roche). The mixture was incubated at 30 °C for 45 min to enable phosphorylation and the reaction was terminated by ultracentrifugation at 100,000$g$ for 30 min. The membrane was then solubilized in 0.5% (w/v) lauryl maltose neopentyl glycol (LMNG; Anatrace), 0.05% (w/v) cholesterol hemisuccinate (CHS; Sigma), 20 mM HEPES (pH 7.5), 125 mM NaCl, 5 mM $MgCl_2$, 30 μM glucagon and 10 μM diC8-PtdIns(4,5)$P_2$ at 4 °C for 4 h. The insoluble debris was removed by ultracentrifugation at 100,000$g$ for 30 min. The supernatant was supplemented with 2 mM $CaCl_2$ and incubated with anti-protein C affinity matrix (Roche) at 4 °C overnight.

The resin was washed with 20 column volumes of washing buffer containing 20 mM HEPES (pH 7.5), 125 mM NaCl, 5 mM $MgCl_2$, 2 mM $CaCl_2$, 1 μM glucagon, 1 μM diC8-PtdIns(4,5)$P_2$, 0.01% (w/v) LMNG, 0.0033% (w/v) glycol-diosgenin (GDN; Anatrace) and 0.001% (w/v) CHS. The complex was eluted with 5 column volumes of elute buffer containing 20 mM HEPES (pH 7.5), 125 mM NaCl, 5 mM EGTA, 0.01% (w/v) LMNG, 0.0033% (w/v) GDN, 0.001% (w/v) CHS, 30 μM glucagon and 50 μM diC8-PtdIns(4,5)$P_2$, and further purified by incubating with TALON Superflow resin (Clontech) at 4 °C for 4 h. The resin was then washed with the washing buffer supplemented with 5 mM imidazole, and the complex was eluted with the elute buffer supplemented with 200 mM imidazole.

The glucagon–GCGR($V_2RC$)–βarr1 complex sample was incubated with Nb32 at a molar ratio of 1:10, and then subjected to size-exclusion chromatography on a Superdex 200 Increase 10/300 column, which was pre-equilibrated with running buffer containing 20 mM HEPES (pH 7.5), 125 mM NaCl, 5 mM $MgCl_2$, 1 μM glucagon, 1 μM diC8-PtdIns(4,5) $P_2$, 0.002% (w/v) LMNG, 0.00067% (w/v) GDN and 0.0002% (w/v) CHS. The peak fractions containing the complex were collected and concentrated to 3 mg ml$^{-1}$ using a 100-kDa molecular weight cut-off concentrator (Millipore) and then analysed by analytical size-exclusion chromatography.

## Cryo-EM sample preparation and data acquisition

Of the glucagon–GCGR($V_2RC$)–βarr1 protein sample, 3 μl was applied to glow-discharged holey grid (ANTcryo R1.2/1.3, Au 300 mesh) and flash frozen in liquid ethane using a Mark IV Vitrobot (Thermo Fisher Scientific) with a blot time of 1.5 s and a blot force of 0 at 4 °C and 100% humidity. Data collection was conducted on a 300 kV Titan Krios G3 electron microscope (FEI) equipped with a Gatan K3 summit direct detection camera and a GIF-Quantum energy filter at a magnification of ×81,000. The movies were captured with a bin2 pixel size of 1.071 Å using the super-resolution counting mode of SerialEM[53]. The defocus values of movies varied from −0.8 to −1.5 μm and the exposure time was a total of 3 s for 40 frames. The dose rate was 1.75 electrons per Å$^2$ per frame.

## Cryo-EM data processing and model building

A total of 5,583 movies were collected and subjected to beam-induced motion correction using MotionCor2 (ref. 54). The contrast transfer function parameters of each micrograph were estimated using CTFFIND4 in CryoSPARC[55]. The following data processing procedures were also performed by CryoSPARC[55]. The particles from 500 micrographs were picked by blob picker and extracted for two rounds of 2D classification. After manual selection, 190,906 particles were subjected to ab initio reconstruction and the projections of the resulting map served as a template to pick particles from the entire dataset. In total, 4,041,891 particles were picked and extracted for 2D classification. The best-looking classes of 2,531,077 particles were subjected to ab initio reconstruction for initial 3D classification, generating five classes of initial models without any preset templates. The particles in the best-looking class were subjected to further 2D classification, ab initio reconstruction and heterogeneous refinement. After removing the class of blurry particles, 551,645 particles were subjected to 3D classification without alignments by setting the number of classes to ten. Two sets of particles were classified, including one in the ligand-bound state (300,738 particles) and the other in the ligand-free state (250,907 particles). These two datasets were subjected to non-uniform refinement and local refinement using a mask encompassing the receptor and βarr1, resulting in two final maps with global resolutions at 3.3 Å and 3.5 Å, respectively. The reported resolution was determined using gold-standard Fourier shell correlation with the 0.143 criteria. Local resolution estimation was determined using ResMap[53]. The 3D variability analysis of the cryo-EM data was performed using 3D variability implemented in CryoSPARC[55] to visualize the dynamics of the glucagon-binding regions in the βarr1-bound and $G_s$-bound complexes of GCGR. The 3D variability analysis was processed using the particles in the final round of non-uniform refinement in data processing.

The initial models of the GCGR($V_2RC$)–βarr1 complexes were built by docking GCGR from the glucagon–GCGR–$G_s$ structure (PDB ID:

6LMK) and the βarr1 from the NTSR1–βarr1 structure (PDB ID: 6UP7) into the maps using Chimera[56]. The phospholipid diC8-PtdIns(4,5)$P_2$ was introduced into both models according to the maps. The models were manually adjusted in Coot 0.8.9 (ref. 57) and refined by several rounds of real-space refinement in PHENIX[58]. The final models were validated using MolProbity[59]. The figures were prepared using Chimera or PyMOL (https://pymol.org/2/).

## BRET assays

To measure the interaction between GCGR and βarr1, the Rluc8 donor and GFP2 acceptor were added to the C termini of the wild-type GCGR (or mutants) and human βarr1, respectively. For measurements of $G_s$ activation, TRUPATH biosensors[60] were used, with the Rluc8 and GFP2 fused to the residue 122 in $G\alpha_s$ and the N terminus of $G\gamma_9$, respectively. Of HEK293F cells (Invitrogen; cells were routinely tested for mycoplasma contamination) at a density of $1.2 \times 10^6$ cells per ml, 2 ml was co-transfected with the plasmids of the above constructs at a ratio of 1:1 (GCGR:βarr1) for the GCGR–βarr1 interaction assay or 1:1:1:1 ($GCGR:G\alpha_s:G\beta_3:G\gamma_9$) for the $G_s$ activation assay, with a total plasmid amount of 4 µg. After 48 h post-transfection, the cell-surface expression of GCGR was measured by detecting the fluorescence signal on the cell surface using a monoclonal anti-FLAG M2-FITC antibody (Sigma; 1:120 diluted in TBS supplemented with 4% BSA and 20% viability staining solution 7-AAD (Invitrogen)) targeting the FLAG tag in the N terminus of GCGR using a flow cytometry reader (Guava easyCyte HT, Millipore) with the software GuavaSoft 3.1. The cells were then plated into 96-well plates at a cell density of 30,000 cells per well in 60 µl of the assay buffer containing 1× Hanks' balanced salt solution (HBSS; Gibco) and 20 mM HEPES (pH 7.4). After a 30-min incubation at 37 °C, 10 µl of freshly prepared 50 µM coelenterazine 400a (Nanolight Technologies) was added into the plates in dark. After incubation for 5 min, the baseline was read by the Synergy II (Bio-Tek) plate reader with 410 nm (Rluc8-coelenterazine 400a) and 515 nm (GFP2) emission filters, at integration times of 3 s per well. Then, 30 µl of glucagon at different concentrations (1 pM to 10 µM; diluted by PBS and 20 mM HEPES (pH 7.4)) was added into each well, and the signals were measured four times in 16 min.

The βarr recruitment at the plasma membrane and GPCR–βarr endocytosis were measured using a bystander BRET approach. For the membrane recruitment, rGFP was fused to the N terminus of the CAAX membrane anchor motif from the human KRAS protein (GKKKKKKKSKTKCVIM) (rGFP–CAAX). For endocytosis, GFP2 was added to the C terminus of the human Endofin FYVE domain (residues Q739–K806) (GFP2–FYVE). The N terminus of the human βarr2 was connected with Rluc8 through a flexible linker (GSSSSG) (Rluc8–βarr2). All the constructs were cloned into the PTT5 vector. The plasmid of the wild-type GCGR (or mutants), $AT_1R$ or $\beta_2AR$ was transiently co-transfected with the plasmids encoding rGFP–CAAX (or GFP2–FYVE) and Rluc8–βarr2 at a ratio of 2:2:1 in 2 ml HEK293F cells at a density of $1.2 \times 10^6$ cells per ml. Protein expression and receptor surface expression measurements were performed as described above. The cells were plated into 96-well white plates (30,000 cells per well) in 60 µl of assay buffer and incubated at 37 °C for 10 min, followed by addition of 10 µl of freshly prepared 50 µM coelenterazine 400a and equilibration for 8 min. The BRET baselines were then measured by the plate reader with 410-nm and 515-nm emission filters for 20 min. Next, 30 µl of ligand at different concentrations (10 pM to 100 µM or 1 pM to 10 µM diluted in 1× HBSS salt solution and 20 mM HEPES (pH 7.4)) were added to each well and the BRET signals were monitored continuously five times.

To validate and measure the GCGR-promoted interaction between βarr1 and $G_s$, the BRET assay was performed as previously described[49]. In brief, βarr1 was modified by adding GFP2 to its C terminus (βarr1–GFP2) and the $G\alpha_s$ subunit was tagged with Rluc8 at position 122 ($G\alpha_s$–Rluc8). The plasmids of βarr1–GFP2, $G\alpha_s$–Rluc8, $G\beta_3$, $G\gamma_9$ and the wild-type

GCGR (or mutants) were co-transfected into 2 ml HEK293F cells at a ratio of 4:2:2:2:1 with a total amount of 4 µg. After 48 h of expression, the cell-surface expression of the receptor was measured as described above, and then the cells were plated into 96-well plates in 60 µl of assay buffer. After 30 min of incubation at 37 °C, 10 µl of freshly prepared 50 µM coelenterazine 400a was added into each well. Then, the baseline was read after a 10-min equilibration. Different concentrations of glucagon (1 pM to 10 µM) were added into the wells to stimulate the co-binding of $G_s$ and βarr1 to GCGR and the signals were read after a 5-min incubation.

All the BRET data were analysed using GraphPad Prism 8.0.

## Reporting summary

Further information on research design is available in the Nature Portfolio Reporting Summary linked to this article.

## Data availability

Atomic coordinates and cryo-EM density maps for the structures of GCGR(V₂RC)–βarr1 and glucagon–GCGR(V₂RC)–βarr1 complexes have been deposited in the Protein Data Bank under identification codes 8JRU and 8JRV, respectively, and in the Electron Microscopy Data Bank under accession codes EMD-36606 and EMD-36607, respectively. The database used in this study includes Protein Data Bank 4ZWJ, 5XEZ, 6LMK, 6U1N, 6UP7, 6TKO, 7R0C and 7SRS.

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

**Acknowledgements** The cryo-EM studies were performed at the electron microscopy facility of the Shanghai Institute of Materia Medica (SIMM), Chinese Academy of Sciences. We thank Q. Wang for cryo-EM data collection. This work was supported by the National Science Foundation of China grant 31825010, National Key R&D Program of China 2022YFA1302900 (to B.W. and S.H.), National Science Foundation of China grant 82121005 (to B.W.), CAS Strategic Priority Research Program XDB37030100 (to B.W. and Q.Z.) and the Shanghai Pilot Program for Basic Research–Chinese Academy of Sciences, Shanghai Branch JCYJ-SHFY-2021-008 (to B.W.).

**Author contributions** K.C. developed the protein expression and purification procedures, prepared protein samples for cryo-EM studies, collected cryo-EM data, performed the cryo-EM data processing and analysis, model building and structure refinement, performed the functional assays and helped with manuscript preparation. C.Z. and S.L. performed the functional assays and helped with data analysis. X.Y., H.C. and Y.L. helped with protein preparation and the functional assays. C.Y., L.M. and X.C. expressed the proteins. Y.Z. and S.H. helped with structure determination and data analysis. Q.Z. and B.W. initiated the project, planned and analysed experiments, supervised the research and wrote the manuscript with input from all co-authors.

**Competing interests** The authors declare no competing interests.

**Additional information**
**Correspondence and requests for materials** should be addressed to Qiang Zhao or Beili Wu.

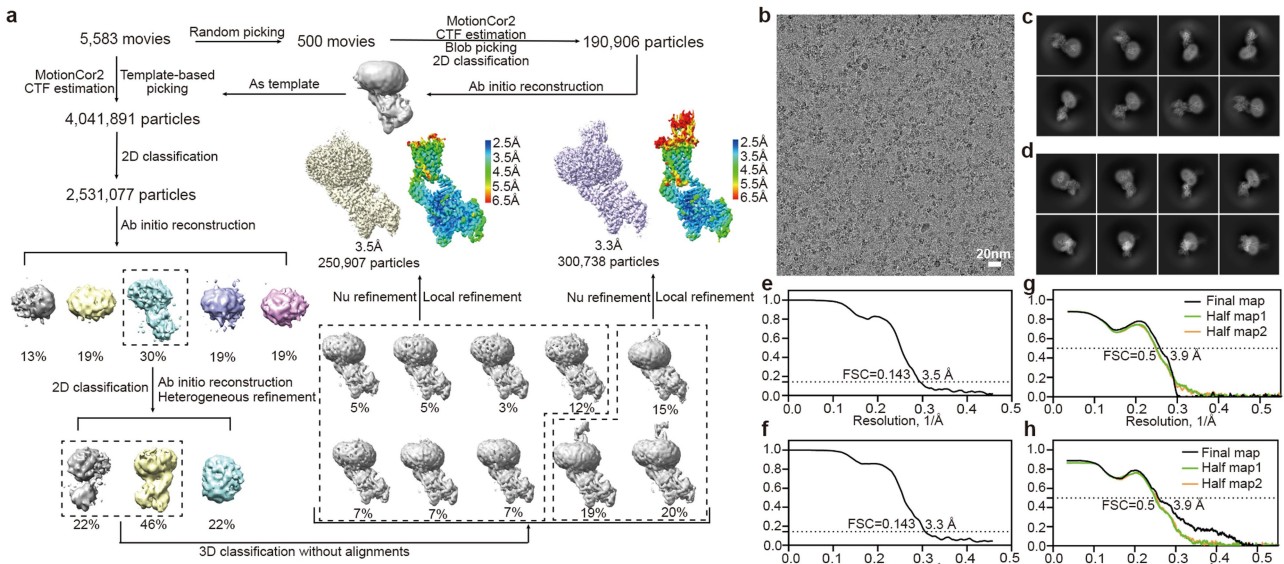

**Extended Data Fig. 1 | Cryo-EM processing and 3D reconstruction workflow.** **a**, Data processing workflow, with cryo-EM maps coloured according to local resolution (in Å). **b**, Representative cryo-EM image from two independent experiments. **c**, **d**, 2D averages of GCGR(V₂RC)−βarr1 (**c**) and glucagon−GCGR(V₂RC)−βarr1 (**d**). **e**, **f**, Gold-standard FSC curves of GCGR(V₂RC)−βarr1 (**e**) and glucagon−GCGR(V₂RC)−βarr1 (**f**). **g**, **h**, Cross-validation of model to cryo-EM density map for GCGR(V₂RC)−βarr1 (**g**) and glucagon−GCGR(V₂RC)−βarr1 (**h**). FSC curves for the final model versus the final map and half maps are shown in black, green and yellow, respectively.

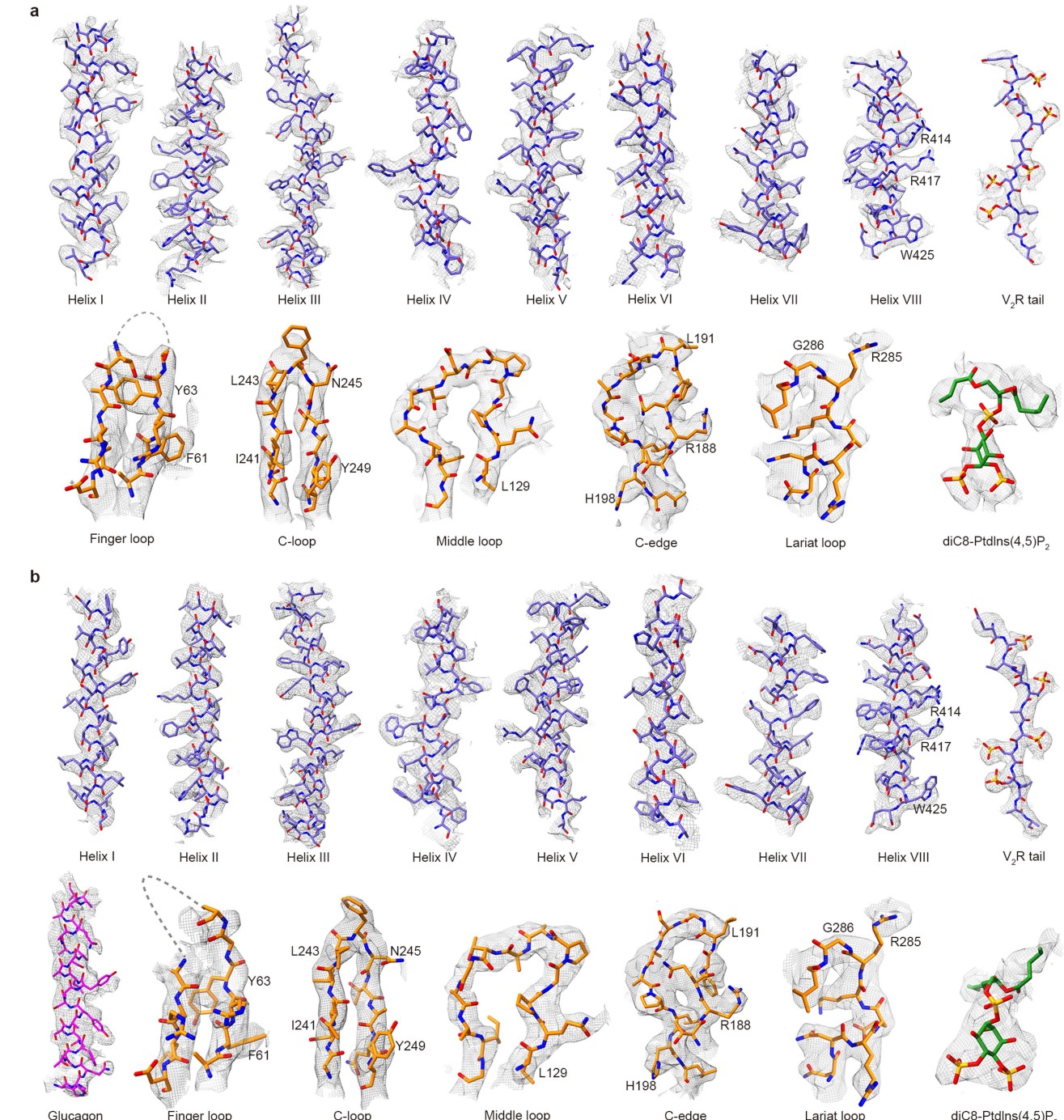

**Extended Data Fig. 2 | Cryo-EM density maps of the GCGR(V₂RC)–βarr1 structures. a**, GCGR(V₂RC)–βarr1; **b**, glucagon–GCGR(V₂RC)–βarr1. Cryo-EM maps and models of the two structures are shown for all transmembrane helices of the receptor, the central loops and C-edge of βarr1, and the phospholipid diC8-PtdIns(4,5)P₂. The models are shown as sticks. The maps are coloured grey.

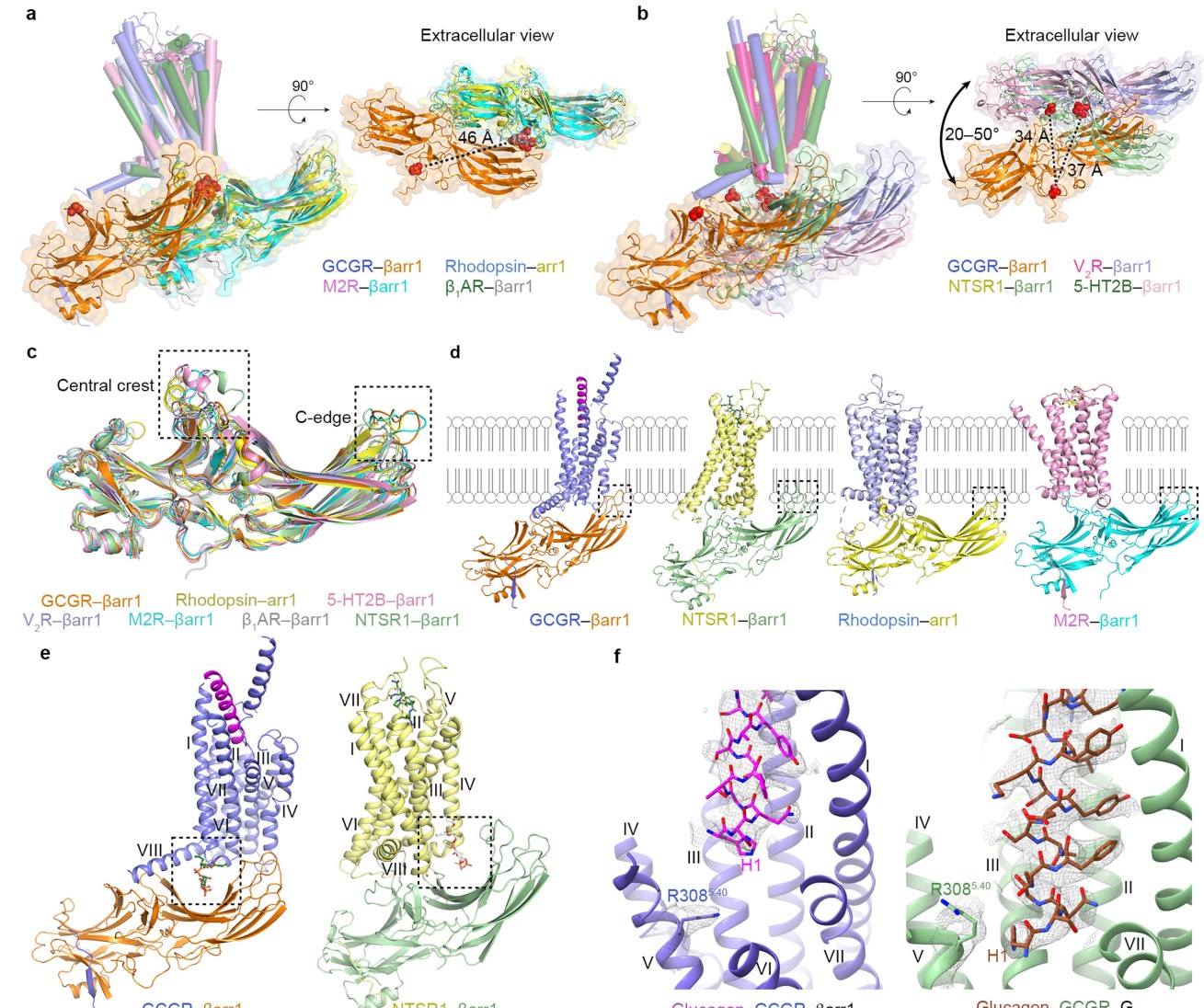

**Extended Data Fig. 3 | Comparison of the glucagon–GCGR(V₂RC)–βarr1 structure with the other known GPCR–arrestin structures and the glucagon–GCGR–Gₛ structure. a**, Structural comparison of the glucagon–GCGR(V₂RC)–βarr1 complex with the rhodopsin–arr1, M2R–βarr1 and β₁AR–βarr1 complexes (PDB IDs: 4ZWJ, 6U1N, and 6TKO). The structures are shown in membrane view (left) and extracellular view (right). Only the arrestins are shown in the extracellular view. The residue D135 in the middle loop of βarr1 (D139 in arr1) in each structure is also shown as red spheres. The distance between the residue D135 in the glucagon–GCGR(V₂RC)–βarr1 structure and the counterparts in the other GPCR–arrestin structures is indicated by a black dashed line. **b**, Structural comparison of the glucagon–GCGR(V₂RC)–βarr1 complex with the V₂R–βarr1, NTSR1–βarr1 and 5-HT2B–βarr1 complexes (PDB IDs: 7R0C, 6UP7, and 7SRS). **c**, Conformational comparison of the arrestins in

the known GPCR–arrestin structures. The central crest and C-edge are highlighted by two black dashed boxes. **d**, Comparison of the arrestin C-edge–membrane interaction in the structures of glucagon–GCGR(V₂RC)–βarr1, NTSR1–βarr1, rhodopsin–arr1 and M2R–βarr1. The C-edge in each structure is highlighted by a black dashed box. **e**, Comparison of the phospholipid binding sites in the glucagon–GCGR(V₂RC)–βarr1 and NTSR1–βarr1 structures. The phospholipid diC8-PtdIns(4,5)P₂ is shown as sticks and its binding site is highlighted by a black dashed box in each structure. **f**, Comparison of the glucagon binding modes and the conformations of the GCGR residue R308⁵·⁴⁰ in the glucagon–GCGR(V₂RC)–βarr1 and glucagon–GCGR–Gₛ structures. Glucagon and the residue R308⁵·⁴⁰ in the two structures are shown as sticks, and their densities are coloured grey.

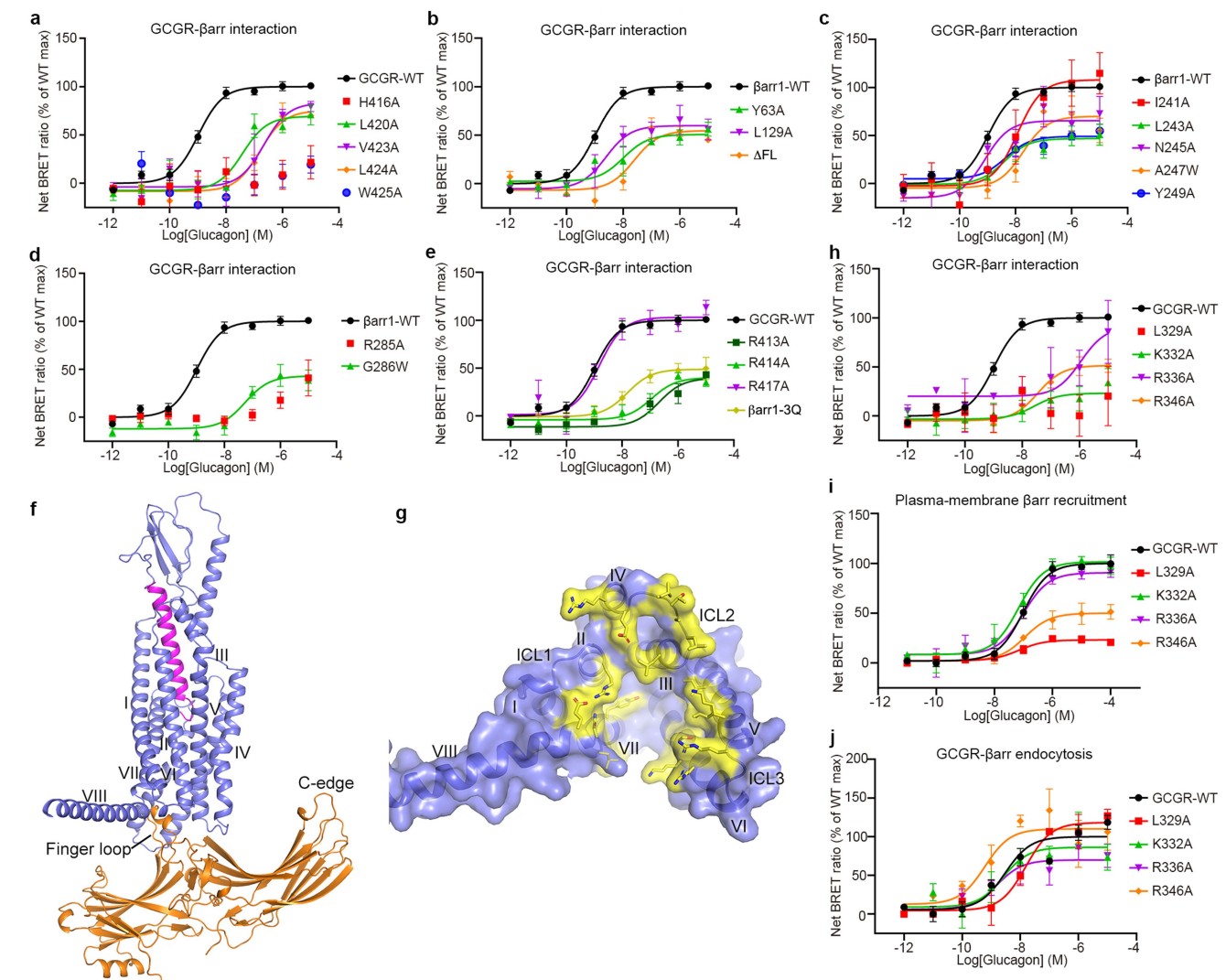

**Extended Data Fig. 4 | Functional assays and mutation design for the potential core conformation. a-e**, Glucagon-induced GCGR-βarr interaction measured by BRET assay. Data are shown as mean ± s.e.m. from at least three independent experiments performed in technical duplicate. Extended Data Table 1 provides detailed numbers of independent experiments (*n*), statistical evaluation, *P* values and expression levels. **a**, Glucagon-induced GCGR-βarr interaction for the wild-type GCGR (WT) and its mutants in helix VIII. **b–d**, Glucagon-induced GCGR-βarr interaction for the wild-type βarr1 (WT) and its mutants. ΔFL, the βarr1 mutant with the turn region of the finger loop (residues 66–73) removed. **e**, Glucagon-induced GCGR-βarr interaction for the wild-type GCGR (WT) and its mutants as well as the 3Q mutant of βarr1. **f**, A model of the GCGR–βarr1 complex in the core conformation. The model

was predicted using Alphafold-Multimer[61]. In brief, the sequences of human GCGR, βarr1 and glucagon were subjected to Alphafold2 to generate a series of complex models following the instruction, by setting num_predicted_model to 5. The models with no physiological significance were excluded. **g**, Intracellular view of the receptor in the predicted model of core conformation. Twenty single mutations of GCGR were designed based on the interaction between GCGR and βarr1 in the model. The residues that were mutated are coloured yellow. **h–j**, The GCGR-βarr interaction, plasma-membrane βarr recruitment, and endocytosis measured by BRET assays. Data are shown as mean ± s.e.m. from at least three independent experiments performed in technical duplicate. Extended Data Tables 1 and 3 provide detailed numbers of independent experiments (*n*), statistical evaluation, *P* values and expression levels.

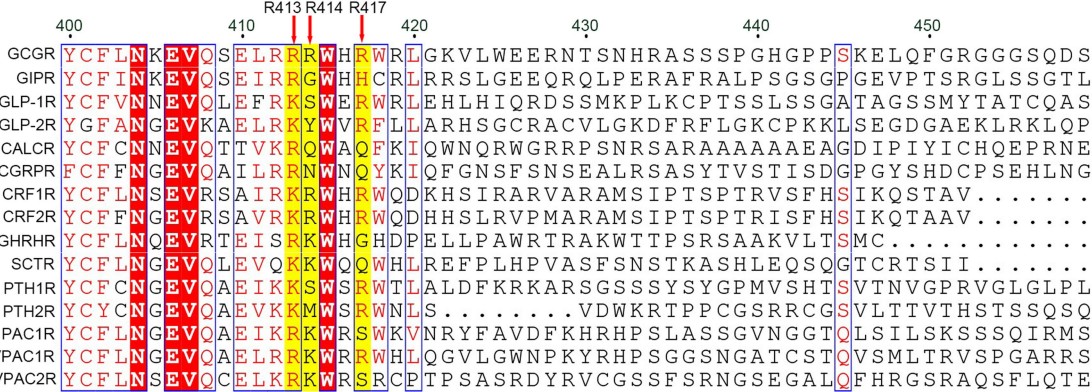

**Extended Data Fig. 5 | Sequence alignment of the C termini of the receptors in the secretin receptor family.** The residues R413, R414 and R417 in GCGR and their counterparts in the other receptors are highlighted by yellow backgrounds. The alignment was generated using UniProt (http://www.uniprot.org/align/) and the graphic was prepared on the ESPript 3.0 server (http://espript.ibcp.fr/ESPript/cgi-bin/ESPript.cgi).

**Extended Data Table 1 | Interaction between GCGR and βarr1, measured by BRET assay**

| | Mutants[†] | $EC_{50}$ (nM) | $EC_{50}$ ratio[‡] | $pEC_{50}$ mean ± s.e.m.[§] | $pEC_{50}$ P value | $E_{max}$[‖] mean ± s.e.m.[§] | $E_{max}$ P value | n[¶] | Expression[#] % of WT | Expression[#] P value |
|---|---|---|---|---|---|---|---|---|---|---|
| | WT | 1.0 | 1 | 8.98 ± 0.09 | / | 100 ± 4 | / | 15 | 100 | / |
| | Construct 1[††] | 3.8 | 4 | 8.42 ± 0.21 | 0.8666 | 74 ± 6 | 0.1339 | 5 | 136 ± 20* | 0.0121 |
| | G165$^{ICL1}$W | 0.30 | 0.3 | 9.53 ± 0.35 | 0.9500 | 80 ± 11 | 0.7197 | 4 | 75 ± 9 | 0.4356 |
| | S167$^{ICL1}$A | 1.7 | 2 | 8.76 ± 0.29 | 0.9994 | 85 ± 10 | 0.9927 | 3 | 71 ± 9 | 0.3789 |
| | R413$^{VIII}$A | 205 | 205 | 6.69 ± 0.29*** | <0.0001 | 51 ± 7*** | <0.0001 | 4 | 77 ± 9 | 0.5908 |
| | R414$^{VIII}$A | 112 | 112 | 6.95 ± 0.27*** | <0.0001 | 44 ± 6*** | <0.0001 | 6 | 80 ± 10 | 0.5669 |
| | H416$^{VIII}$A | nd | nd | nd | nd | nd | nd | 5 | 60 ± 11* | 0.0029 |
| | R417$^{VIII}$A | 1.5 | 2 | 8.83 ± 0.21 | 0.9996 | 102 ± 9 | 0.9998 | 4 | 73 ± 9 | 0.3023 |
| | L420$^{VIII}$A | 38 | 38 | 7.42 ± 0.24** | 0.0002 | 77 ± 8 | 0.4570 | 4 | 76 ± 9 | 0.5117 |
| | V423$^{VIII}$A | 219 | 219 | 6.66 ± 0.37*** | <0.0001 | 87 ± 15 | 0.9985 | 3 | 79 ± 9 | 0.8894 |
| | L424$^{VIII}$A | 161 | 161 | 6.79 ± 0.21*** | <0.0001 | 84 ± 8 | 0.9576 | 4 | 67 ± 10 | 0.0744 |
| | W425$^{VIII}$A | nd | nd | nd | nd | nd | nd | 3 | 70 ± 10 | 0.3208 |
| GCGR | R173$^{2.46}$A | 2.3 | 2 | 8.64 ± 0.36 | 0.9986 | 68 ± 10* | 0.0176 | 5 | 105 ± 12 | 0.9995 |
| | L249$^{3.54}$A | 1.0 | 1 | 9.00 ± 0.22 | >0.9999 | 93 ± 9 | 0.9993 | 4 | 19 ± 3*** | <0.0001 |
| | L252$^{3.57}$A | 1.9 | 2 | 8.71 ± 0.26 | 0.9993 | 98 ± 10 | 0.9999 | 3 | 40 ± 8** | 0.0002 |
| | L255$^{ICL2}$A | 0.57 | 0.6 | 9.24 ± 0.45 | 0.9993 | 57 ± 11* | 0.0052 | 3 | 92 ± 13 | 0.9993 |
| | A256$^{ICL2}$W | 0.73 | 0.7 | 9.14 ± 0.18 | 0.9996 | 92 ± 7 | 0.9993 | 3 | 101 ± 20 | >0.9999 |
| | T257$^{ICL2}$A | 1.5 | 2 | 8.82 ± 0.18 | 0.9996 | 100 ± 8 | >0.9999 | 4 | 80 ± 10 | 0.8146 |
| | L258$^{ICL2}$A | 1.0 | 1 | 9.00 ± 0.26 | >0.9999 | 64 ± 7* | 0.0450 | 3 | 88 ± 9 | 0.9988 |
| | E260$^{ICL2}$A | 1.5 | 2 | 8.84 ± 0.24 | 0.9997 | 129 ± 13 | 0.2540 | 4 | 67 ± 8 | 0.1829 |
| | R261$^{ICL2}$A | 0.47 | 0.5 | 9.33 ± 0.27 | 0.9988 | 64 ± 7* | 0.0119 | 4 | 88 ± 7 | 0.9986 |
| | I325$^{5.57}$A | 0.71 | 0.7 | 9.15 ± 0.23 | 0.9996 | 88 ± 9 | 0.9987 | 3 | 62 ± 10 | 0.0599 |
| | L329$^{5.61}$A | nd | nd | nd | nd | nd | nd | 4 | 46 ± 12*** | <0.0001 |
| | K332$^{5.64}$A | 27 | 27 | 7.58 ± 0.80* | 0.0016 | 26 ± 9*** | <0.0001 | 4 | 58 ± 6* | 0.0050 |
| | R336$^{ICL3}$A | 1,122 | 1,122 | 5.95 ± 0.58*** | <0.0001 | 71 ± 28 | 0.2540 | 3 | 47 ± 8** | 0.0008 |
| | R346$^{6.37}$A | 36 | 36 | 7.45 ± 0.21* | 0.0025 | 56 ± 5* | 0.0037 | 3 | 73 ± 9 | 0.5098 |
| | K349$^{6.40}$A | 1.7 | 2 | 8.77 ± 0.25 | 0.9994 | 83 ± 8 | 0.9796 | 3 | 97 ± 11 | 0.9998 |
| | S350$^{6.41}$A | 1.2 | 1 | 8.92 ± 0.24 | 0.9999 | 75 ± 8 | 0.5222 | 3 | 77 ± 9 | 0.7839 |
| | Y400$^{7.57}$A | 5.4 | 5 | 8.27 ± 0.21 | 0.6362 | 49 ± 4*** | <0.0001 | 4 | 44 ± 7*** | <0.0001 |
| | L403$^{7.60}$A | 1.9 | 2 | 8.73 ± 0.30 | 0.9993 | 45 ± 6*** | <0.0001 | 3 | 61 ± 7* | 0.0468 |
| | N404$^{7.61}$A | 7.5 | 8 | 8.13 ± 0.21 | 0.5149 | 79 ± 7 | 0.8224 | 3 | 59 ± 7* | 0.0280 |
| | E406$^{7.63}$A | 0.29 | 0.3 | 9.53 ± 0.23 | 0.9842 | 106 ± 10 | 0.9995 | 3 | 47 ± 3** | 0.0008 |
| βarr1 | Construct 2[‡‡] | 2.5 | 3 | 8.61 ± 0.22 | 0.9986 | 79 ± 7 | 0.6328 | 4 | / | / |
| | Y63A | 10 | 10 | 7.99 ± 0.32 | 0.1044 | 48 ± 7*** | <0.0001 | 4 | / | / |
| | L129A | 2.5 | 3 | 8.61 ± 0.33 | 0.9985 | 65 ± 9* | 0.0055 | 5 | / | / |
| | I241A | 12 | 12 | 7.93 ± 0.31 | 0.1636 | 110 ± 15 | 0.9990 | 3 | / | / |
| | L243A | 4.0 | 4 | 8.40 ± 0.35 | 0.9776 | 48 ± 7** | 0.0002 | 3 | / | / |
| | N245A | 0.82 | 1 | 9.09 ± 0.29 | 0.9997 | 80 ± 10 | 0.7197 | 4 | / | / |
| | A247W | 18 | 18 | 7.74 ± 0.30* | 0.0379 | 75 ± 10 | 0.5222 | 3 | / | / |
| | Y249A | 4.5 | 5 | 8.35 ± 0.25 | 0.6970 | 44 ± 4*** | <0.0001 | 5 | / | / |
| | R285A | nd | nd | nd | nd | nd | nd | 3 | / | / |
| | G286W | 56 | 56 | 7.25 ± 0.28*** | <0.0001 | 55 ± 7** | 0.0004 | 4 | / | / |
| | 3Q | 15 | 15 | 7.82 ± 0.25* | 0.0218 | 49 ± 5*** | <0.0001 | 4 | / | / |
| | ΔFL | 24 | 24 | 7.61 ± 0.30* | 0.0119 | 61 ± 8* | 0.0187 | 3 | / | / |

[†]All mutations were introduced in the wild-type GCGR or βarr1 (WT). ΔFL, the βarr1 mutant with the turn region of the finger loop (residues 66–73) removed.

[‡]The $EC_{50}$ ratio ($EC_{50(mutant)}/EC_{50(WT)}$) represents the shift between the WT and mutant curves, and characterizes the effect of the mutations on the GCGR-βarr1 interaction.

[§]Data are mean ± s.e.m. from at least three independent experiments. *$P<0.05$, **$P<0.001$, ***$P<0.0001$ by one-way analysis of variance followed by Dunnett's post-test compared to the response of WT. nd, not determined (data for which the concentration response curve could not reach effect saturation within the concentration range tested).

[‖]The maximal response is reported as a percentage of the maximum effect at the WT.

[¶]Sample size, the number of independent experiments performed in technical duplicate.

[#]Protein expression levels of GCGR constructs at the cell surface were determined in parallel by flow cytometry with an anti-Flag antibody (Sigma) and reported as per cent compared to the WT from at least three independent measurements performed in technical duplicate.

[††]Construct 1, the GCGR construct that was used to determine the structures.

[‡‡]Construct 2, the βarr1 construct that was used to determine the structures.

**Extended Data Table 2 | Cryo-EM data collection, refinement and validation statistics**

| | Glucagon–GCGR$^{V2RC}$–βarr1 (EMDB-36607) (PDB 8JRV) | GCGR$^{V2RC}$–βarr1 (EMDB-36606) (PDB 8JRU) |
|---|---|---|
| **Data collection and processing** | | |
| Magnification | 81,000 | 81,000 |
| Voltage (kV) | 300 | 300 |
| Electron exposure (e⁻/Å²) | 70 | 70 |
| Defocus range (μm) | −0.8 ~ −1.5 | −0.8 ~ −1.5 |
| Pixel size (Å) | 1.071 | 1.071 |
| Symmetry imposed | C1 | C1 |
| Initial particle images (no.) | 4,041,891 | 4,041,891 |
| Final particle images (no.) | 300,738 | 250,907 |
| Map resolution (Å) | 3.3 | 3.5 |
| FSC threshold | 0.143 | 0.143 |
| Map resolution range (Å) | 2.5–6.5 | 2.5–6.5 |
| | | |
| **Refinement** | | |
| Initial model used (PDB code) | 6LMK and 6UP7 | 6LMK and 6UP7 |
| Model resolution (Å) | 3.9 | 3.9 |
| FSC threshold | 0.5 | 0.5 |
| Map sharpening $B$ factor (Å²) | −57 | −59 |
| Model composition | | |
| Non-hydrogen atoms | 7,707 | 7,120 |
| Protein/peptide residues | 1,036 | 947 |
| $B$ factors (Å²) | | |
| Protein/peptide | 103.68 | 71.89 |
| Lipid | 117.94 | 93.67 |
| R.m.s. deviations | | |
| Bond lengths (Å) | 0.002 | 0.002 |
| Bond angles (°) | 0.576 | 0.548 |
| Validation | | |
| Molprobity score | 1.73 | 1.79 |
| Clashscore | 6.77 | 7.50 |
| Poor rotamers (%) | 0.00 | 0.29 |
| Ramachandran plot | | |
| Favored (%) | 94.83 | 94.52 |
| Allowed (%) | 5.17 | 5.48 |
| Disallowed (%) | 0.00 | 0.00 |

**Extended Data Table 3 | Agonist-induced plasma-membrane βarr2 recruitment and GPCR–βarr2 endocytosis, measured by BRET assay**

| | Mutants[†] | $EC_{50}$ (nM) | $EC_{50}$ ratio[‡] | pEC$_{50}$ mean ± s.e.m.[§] | pEC$_{50}$ P value | $E_{max}$[‖] mean ± s.e.m.[§] | $E_{max}$ P value | n[¶] | Expression[#] % of WT | Expression[#] P value |
|---|---|---|---|---|---|---|---|---|---|---|
| | | | | **Plasma-membrane βarr2 recruitment** | | | | | | |
| GCGR | WT | 99 | 1 | 7.01 ± 0.10 | / | 100 ± 4 | / | 8 | 100 | / |
| | GCGR[V2RC††] | 50 | 0.5 | 7.30 ± 0.14 | 0.9812 | 95 ± 5 | 0.9958 | 5 | 91 ± 2 | 0.9990 |
| | L329[5.61]A | 69 | 0.7 | 7.16 ± 0.23 | 0.9995 | 21 ± 2*** | <0.0001 | 3 | 76 ± 12 | 0.6572 |
| | K332[5.64]A | 68 | 0.7 | 7.17 ± 0.16 | 0.9994 | 101 ± 7 | 0.9999 | 3 | 84 ± 3 | 0.9676 |
| | R336[ICL3]A | 99 | 1 | 7.00 ± 0.14 | >0.9999 | 86 ± 5 | 0.5032 | 3 | 97 ± 4 | 0.9997 |
| | R346[6.37]A | 113 | 1 | 6.95 ± 0.22 | 0.9998 | 49 ± 5*** | <0.0001 | 3 | 89 ± 9 | 0.9989 |
| | R413[VIII]A | 23 | 0.2 | 7.65 ± 0.69 | 0.4893 | 15 ± 4*** | <0.0001 | 3 | 100 ± 4 | >0.9999 |
| | R414[VIII]A | 306 | 3 | 6.51 ± 0.30 | 0.6833 | 67 ± 8** | 0.0002 | 4 | 72 ± 10 | 0.3055 |
| | H416[VIII]A | 72 | 0.7 | 7.14 ± 0.28 | 0.9995 | 29 ± 4*** | <0.0001 | 4 | 65 ± 21 | 0.0924 |
| | R417[VIII]A | 61 | 0.6 | 7.21 ± 0.13 | 0.9993 | 101 ± 6 | 0.9999 | 3 | 85 ± 11 | 0.9815 |
| | L420[VIII]A | 19 | 0.2 | 7.73 ± 0.20 | 0.2161 | 36 ± 3*** | <0.0001 | 4 | 84 ± 15 | 0.9322 |
| | V423[VIII]A | 33 | 0.3 | 7.48 ± 0.25 | 0.7543 | 27 ± 3*** | <0.0001 | 4 | 107 ± 10 | 0.9993 |
| | L424[VIII]A | nd | nd | nd | nd | nd | nd | 3 | 90 ± 17 | 0.9989 |
| | W425[VIII]A | nd | nd | nd | nd | nd | nd | 3 | 93 ± 8 | 0.9994 |
| βarr2 | 3Q | 5.1 | 0.05 | 8.29 ± 0.53* | 0.0055 | 26 ± 6*** | <0.0001 | 3 | / | / |
| β₂AR | WT | 181 | 2 | 6.74 ± 0.13 | 0.9913 | 45 ± 3*** | <0.0001 | 4 | 98 ± 7 | 0.9998 |
| AT₁R | WT | 0.55 | 0.006 | 9.26 ± 0.23*** | <0.0001 | 140 ± 14*** | <0.0001 | 3 | 107 ± 6 | 0.9994 |
| | | | | **GPCR–βarr2 endocytosis** | | | | | | |
| GCGR | WT | 3.0 | 1 | 8.52 ± 0.21 | / | 100 ± 8 | / | 5 | 100 | / |
| | GCGR[V2RC††] | 0.50 | 0.2 | 9.30 ± 0.57 | 0.5127 | 79 ± 19 | 0.8239 | 3 | 86 ± 4 | 0.9104 |
| | L329[5.61]A | 16 | 5 | 7.81 ± 0.20 | 0.6296 | 110 ± 10 | 0.9991 | 3 | 86 ± 3 | 0.9104 |
| | K332[5.64]A | 2.6 | 1 | 8.59 ± 0.40 | 0.9998 | 78 ± 13 | 0.6965 | 4 | 71 ± 5 | 0.0749 |
| | R336[ICL3]A | 1.6 | 0.5 | 8.80 ± 0.34 | 0.9958 | 62 ± 9 | 0.0736 | 5 | 94 ± 7 | 0.9991 |
| | R346[6.37]A | 0.61 | 0.2 | 9.22 ± 0.37 | 0.6465 | 94 ± 15 | 0.9995 | 3 | 98 ± 6 | 0.9997 |
| | R413[VIII]A | 44 | 15 | 7.35 ± 0.27 | 0.0575 | 74 ± 8 | 0.4934 | 4 | 107 ± 2 | 0.9990 |
| | R414[VIII]A | 0.11 | 0.03 | 9.97 ± 0.24* | 0.0212 | 101 ± 11 | >0.9999 | 3 | 64 ± 9* | 0.0276 |
| | H416[VIII]A | 30 | 10 | 7.52 ± 0.26 | 0.2217 | 105 ± 12 | 0.9996 | 4 | 63 ± 6* | 0.0217 |
| | R417[VIII]A | 1.8 | 0.6 | 8.75 ± 0.28 | 0.9993 | 102 ± 12 | 0.9999 | 3 | 116 ± 7 | 0.8123 |
| | L420[VIII]A | nd | nd | nd | nd | nd | nd | 3 | 101 ± 14 | >0.9999 |
| | V423[VIII]A | 25 | 8 | 7.61 ± 0.29 | 0.3224 | 90 ± 11 | 0.9991 | 3 | 107 ± 10 | 0.9991 |
| | L424[VIII]A | 1.6 | 0.5 | 8.81 ± 0.26 | 0.9960 | 80 ± 9 | 0.7929 | 4 | 120 ± 14 | 0.4370 |
| | W425[VIII]A | 148 | 49 | 6.83 ± 0.22* | 0.0048 | 82 ± 9 | 0.9218 | 3 | 103 ± 8 | 0.9997 |
| βarr2 | 3Q | nd | nd | nd | nd | nd | nd | 3 | / | / |
| β₂AR | WT | nd | nd | nd | nd | nd | nd | 3 | 92 ± 16 | 0.9990 |
| AT₁R | WT | 7.7 | 3 | 8.11 ± 0.30 | 0.9807 | 98 ± 13 | 0.9999 | 3 | 110 ± 8 | 0.9903 |

[†]All mutations were introduced in the wild-type GCGR or βarr2 (WT).

[‡]The $EC_{50}$ ratio ($EC_{50(mutant)}$/$EC_{50(WT)}$) represents the shift between the WT and mutant curves, and characterizes the effect of the mutations on the plasma-membrane recruitment or endocytosis.

[§]Data are mean ± s.e.m. from at least three independent experiments. *$P < 0.05$, **$P < 0.001$, ***$P < 0.0001$ by one-way analysis of variance followed by Dunnett's post-test compared to the response of WT. nd, not determined (data for which the concentration response curve could not reach effect saturation within the concentration range tested).

[‖]The maximal response is reported as a percentage of the maximum effect at the WT.

[¶]Sample size, the number of independent experiments performed in technical duplicate.

[#]Protein expression levels of GCGR, β₂AR and AT₁R constructs at the cell surface were determined in parallel by flow cytometry with an anti-Flag antibody (Sigma) and reported as per cent compared to the wild-type GCGR from at least three independent measurements performed in technical duplicate.

[††]The GCGR construct with the C-terminal region (residues H433–F477) replaced with the C-terminal tail of $V_2R$ (residues A343–S371).

**Extended Data Table 4 | Glucagon-induced interaction between βarr1 and G_s at GCGR and glucagon-induced G_s activation of GCGR, measured by BRET assays**

**Interaction between βarr1 and G_s**

| | Mutants[†] | $EC_{50}$ (nM) | $EC_{50}$ ratio[‡] | pEC_{50} mean ± s.e.m.[§] | pEC_{50} P value | $E_{max}$[‖] mean ± s.e.m.[§] | $E_{max}$ P value | n[¶] | Expression[#] % of WT | Expression P value |
|---|---|---|---|---|---|---|---|---|---|---|
| GCGR | WT | 4.9 | 1 | 8.31 ± 0.10 | / | 100 ± 4 | / | 13 | 100 | / |
| | GCGR^V2RC†† | 5.2 | 1 | 8.28 ± 0.38 | >0.9999 | 48 ± 8* | 0.0041 | 3 | 49 ± 11* | 0.0183 |
| | R413VIIIA | 36 | 7 | 7.45 ± 0.20 | 0.5151 | 149 ± 13* | 0.0080 | 3 | 106 ± 19 | 0.9996 |
| | R414VIIIA | 35 | 7 | 7.45 ± 0.30 | 0.5151 | 81 ± 9 | 0.7923 | 3 | 50 ± 16* | 0.0219 |
| | H416VIIIA | 111 | 23 | 6.95 ± 0.24* | 0.0037 | 66 ± 8* | 0.0136 | 7 | 76 ± 10 | 0.2952 |
| | R417VIIIA | 1,032 | 211 | 5.99 ± 0.35*** | <0.0001 | 88 ± 19 | 0.9674 | 4 | 55 ± 10* | 0.0203 |
| | L420VIIIA | 51 | 10 | 7.29 ± 0.25 | 0.0785 | 95 ± 11 | 0.9994 | 6 | 82 ± 14 | 0.7119 |
| | V423VIIIA | 37 | 8 | 7.43 ± 0.77 | 0.3416 | 35 ± 11*** | <0.0001 | 4 | 87 ± 13 | 0.9756 |
| | L424VIIIA | 42 | 9 | 7.38 ± 0.25 | 0.1374 | 70 ± 8 | 0.0594 | 6 | 82 ± 12 | 0.7119 |
| | W425VIIIA | 77 | 16 | 7.12 ± 0.64* | 0.0411 | 28 ± 8*** | <0.0001 | 5 | 92 ± 14 | 0.9971 |
| β_2AR | WT | nd | nd | nd | nd | nd | nd | 4 | 153 ± 12*** | <0.0001 |

**G_s activation**

| | Mutants[†] | $EC_{50}$ (nM) | $EC_{50}$ ratio[‡] | pEC_{50} mean ± s.e.m.[§] | pEC_{50} P value | $E_{max}$[‖] mean ± s.e.m.[§] | $E_{max}$ P value | n[¶] | Expression[#] % of WT | Expression P value |
|---|---|---|---|---|---|---|---|---|---|---|
| GCGR | WT | 10 | 1 | 8.00 ± 0.11 | / | 100 ± 5 | / | 8 | 100 | / |
| | GCGR^V2RC†† | 13 | 1 | 7.89 ± 0.22 | 0.9996 | 87 ± 8 | 0.8724 | 3 | 44 ± 11*** | <0.0001 |
| | R413VIIIA | 16 | 2 | 7.79 ± 0.27 | 0.9928 | 72 ± 8* | 0.0333 | 5 | 102 ± 10 | 0.9997 |
| | R414VIIIA | 14 | 1 | 7.86 ± 0.17 | 0.9995 | 99 ± 7 | 0.9999 | 3 | 95 ± 24 | 0.9994 |
| | H416VIIIA | 17 | 2 | 7.78 ± 0.22 | 0.9916 | 65 ± 6* | 0.0040 | 5 | 97 ± 7 | 0.9996 |
| | R417VIIIA | 15 | 2 | 7.82 ± 0.19 | 0.9961 | 74 ± 6* | 0.0278 | 7 | 102 ± 8 | 0.9997 |
| | L420VIIIA | 15 | 2 | 7.83 ± 0.29 | 0.9971 | 74 ± 9 | 0.0575 | 5 | 92 ± 3 | 0.9729 |
| | V423VIIIA | 32 | 3 | 7.50 ± 0.23 | 0.4043 | 74 ± 7* | 0.0278 | 7 | 97 ± 4 | 0.9996 |
| | L424VIIIA | 25 | 3 | 7.60 ± 0.32 | 0.8929 | 71 ± 9 | 0.0866 | 3 | 96 ± 5 | 0.9996 |
| | W425VIIIA | 12 | 1 | 7.92 ± 0.21 | 0.9997 | 72 ± 6* | 0.0333 | 5 | 99 ± 3 | 0.9999 |

[†]All mutations were introduced in the wild-type GCGR (WT).

[‡]The $EC_{50}$ ratio ($EC_{50(mutant)}/EC_{50(WT)}$) represents the shift between the WT and mutant curves, and characterizes the effect of the mutations on the interaction between βarr1 and G_s or G_s activation.

[§]Data are mean ± s.e.m. from at least three independent experiments. *$P < 0.05$, **$P < 0.001$, ***$P < 0.0001$ by one-way analysis of variance followed by Dunnett's post-test compared to the response of WT. nd, not determined (data for which the concentration response curve could not reach effect saturation within the concentration range tested).

[‖]The maximal response is reported as a percentage of the maximum effect at the WT.

[¶]Sample size, the number of independent experiments performed in technical duplicate.

[#]Protein expression levels of GCGR and β_2AR constructs at the cell surface were determined in parallel by flow cytometry with an anti-Flag antibody (Sigma) and reported as per cent compared to the WT from at least three independent measurements performed in technical duplicate.

[††]The GCGR construct with the C-terminal region (residues H433–F477) replaced with the C-terminal tail of V_2R (residues A343–S371).

# Reporting Summary

## Statistics

For all statistical analyses, confirm that the following items are present in the figure legend, table legend, main text, or Methods section.

| n/a | Confirmed | |
|---|---|---|
| ☐ | ☒ | The exact sample size (*n*) for each experimental group/condition, given as a discrete number and unit of measurement |
| ☐ | ☒ | A statement on whether measurements were taken from distinct samples or whether the same sample was measured repeatedly |
| ☐ | ☒ | The statistical test(s) used AND whether they are one- or two-sided<br>*Only common tests should be described solely by name; describe more complex techniques in the Methods section.* |
| ☒ | ☐ | A description of all covariates tested |
| ☒ | ☐ | A description of any assumptions or corrections, such as tests of normality and adjustment for multiple comparisons |
| ☐ | ☒ | A full description of the statistical parameters including central tendency (e.g. means) or other basic estimates (e.g. regression coefficient) AND variation (e.g. standard deviation) or associated estimates of uncertainty (e.g. confidence intervals) |
| ☐ | ☒ | For null hypothesis testing, the test statistic (e.g. *F*, *t*, *r*) with confidence intervals, effect sizes, degrees of freedom and *P* value noted<br>*Give P values as exact values whenever suitable.* |
| ☒ | ☐ | For Bayesian analysis, information on the choice of priors and Markov chain Monte Carlo settings |
| ☒ | ☐ | For hierarchical and complex designs, identification of the appropriate level for tests and full reporting of outcomes |
| ☒ | ☐ | Estimates of effect sizes (e.g. Cohen's *d*, Pearson's *r*), indicating how they were calculated |

*Our web collection on statistics for biologists contains articles on many of the points above.*

## Software and code

Policy information about availability of computer code

| Data collection | Automated data collection on the Titan Krios was performed using serialEM 3.7. |
|---|---|
| Data analysis | The following softwares were used in cryo-EM data processing, model building, and structure validation: MotionCor2 v1.4.2, CryoSPARC 4.1, ResMap v1.1.4, ChimeraX v.1.1, COOT 0.8.9, PHENIX 1.19.2, and MolProbity 4.2.<br>The functional data were analyzed by GraphPad Prism 8.0.<br>The figures were prepared using PyMOL 1.8 and UCSF Chimera 1.15.<br>The sequence alignment graphic was prepared on the ESPript 3.0 server.<br>The flow cytometry data were collected and analyzed by GuavaSoft 3.1, Guava ExpressPlus panel. |

For manuscripts utilizing custom algorithms or software that are central to the research but not yet described in published literature, software must be made available to editors and reviewers. We strongly encourage code deposition in a community repository (e.g. GitHub). See the Nature Portfolio guidelines for submitting code & software for further information.

# Data

Policy information about availability of data

All manuscripts must include a data availability statement. This statement should provide the following information, where applicable:
- Accession codes, unique identifiers, or web links for publicly available datasets
- A description of any restrictions on data availability
- For clinical datasets or third party data, please ensure that the statement adheres to our policy

> Atomic coordinates and cryo-EM density maps for the structures of GCGRV2RC–barr1 and glucagon–GCGRV2RC–barr1 complexes have been deposited in the Protein Data Bank (PDB) under identification codes 8JRU and 8JRV, respectively, and in the Electron Microscopy Data Bank under accession codes EMD-36606 and EMD-36607, respectively. The database used in this study includes PDB 4ZWJ, 5XEZ, 6LMK, 6U1N, 6UP7, 6TKO, 7R0C, and 7SRS.

# Research involving human participants, their data, or biological material

Policy information about studies with human participants or human data. See also policy information about sex, gender (identity/presentation), and sexual orientation and race, ethnicity and racism.

| | |
|---|---|
| Reporting on sex and gender | No human research participants are involved in this study. |
| Reporting on race, ethnicity, or other socially relevant groupings | No human research participants are involved in this study. |
| Population characteristics | No human research participants are involved in this study. |
| Recruitment | No human research participants are involved in this study. |
| Ethics oversight | No human research participants are involved in this study. |

Note that full information on the approval of the study protocol must also be provided in the manuscript.

# Field-specific reporting

Please select the one below that is the best fit for your research. If you are not sure, read the appropriate sections before making your selection.

☒ Life sciences  ☐ Behavioural & social sciences  ☐ Ecological, evolutionary & environmental sciences

For a reference copy of the document with all sections, see nature.com/documents/nr-reporting-summary-flat.pdf

# Life sciences study design

All studies must disclose on these points even when the disclosure is negative.

| | |
|---|---|
| Sample size | No statistical methods were used to predetermine sample size. All functional data were obtained from at least three independent experiments to ensure each data point was repeatable. Wild-type receptors were tested in parallel as controls with a large number of repeats. The sample sizes of the mutants were evaluated by calculating the standard error of the mean. Sample size for the cryo-EM studies was determined by availability of microscope time and to ensure unambiguous modeling of most of residues that allowed us to obtain a high-resolution reconstruction. |
| Data exclusions | No data were excluded from the analyses. |
| Replication | The functional assays were performed in technical duplicate. All the independent experiments were performed within a month. All attempts at replication were successful. The cryo-EM data were collected from two independent experiments performed within two months. |
| Randomization | Randomization is not relevant to this study, as all experiments did not allocate experimental groups. |
| Blinding | Blinding is not relevant to this study, as no subjective allocation was involved in any of the structural and functional experiments. |

# Reporting for specific materials, systems and methods

We require information from authors about some types of materials, experimental systems and methods used in many studies. Here, indicate whether each material, system or method listed is relevant to your study. If you are not sure if a list item applies to your research, read the appropriate section before selecting a response.

## Materials & experimental systems

| n/a | Involved in the study |
|---|---|
| ☐ | ☒ Antibodies |
| ☐ | ☒ Eukaryotic cell lines |
| ☒ | ☐ Palaeontology and archaeology |
| ☒ | ☐ Animals and other organisms |
| ☒ | ☐ Clinical data |
| ☒ | ☐ Dual use research of concern |
| ☒ | ☐ Plants |

## Methods

| n/a | Involved in the study |
|---|---|
| ☒ | ☐ ChIP-seq |
| ☐ | ☒ Flow cytometry |
| ☒ | ☐ MRI-based neuroimaging |

## Antibodies

| | |
|---|---|
| Antibodies used | ANTI-FLAG M2-FITC antibody: Sigma, Cat#F4049, 1:120 diluted in TBS supplemented with 4% BSA and 20% viability staining solution 7-AAD (Invitrogen). |
| Validation | The ANTI-FLAG M2-FITC antibody was commercially obtained and the validation report is available in the supplier website: https://www.sigmaaldrich.cn/deepweb/assets/sigmaaldrich/product/documents/731/190/f4049dat-mk.pdf; https://www.sigmaaldrich.com/technical-documents/articles/biofiles/antibodies-to-peptides.html. |

## Eukaryotic cell lines

Policy information about cell lines and Sex and Gender in Research

| | |
|---|---|
| Cell line source(s) | The Sf9 and HEK293F cell lines were originally obtained from Invitrogen. |
| Authentication | None of the cell lines have been authenticated. |
| Mycoplasma contamination | The cell lines were negative for mycoplasma contamination. |
| Commonly misidentified lines (See ICLAC register) | No commonly misidentified cell lines were used. |

## Flow Cytometry

### Plots

Confirm that:

☐ The axis labels state the marker and fluorochrome used (e.g. CD4-FITC).

☐ The axis scales are clearly visible. Include numbers along axes only for bottom left plot of group (a 'group' is an analysis of identical markers).

☐ All plots are contour plots with outliers or pseudocolor plots.

☐ A numerical value for number of cells or percentage (with statistics) is provided.

### Methodology

| | |
|---|---|
| Sample preparation | Cell surface expression levels of GCGR and mutants were measured by incubating 10 ul cells with 15 ul monoclonal anti-Flag M2-FITC antibody (Sigma; 1:120 diluted in TBS supplemented with 4% BSA and 20% viability staining solution 7-AAD (Invitrogen)) at 4 °C for 20 min. After incubation, 175 ul TBS buffer was added and the fluorescent signal was measured using a flow cytometry reader (Guava easyCyte HT, Millipore). |
| Instrument | Guava easyCyte HT, Millipore |
| Software | The data were collected and analyzed by GuavaSoft 3.1, Guava ExpressPlus panel. |
| Cell population abundance | For each measurement, 2,000 cell events were collected and the fluorescence intensity of cell population with protein expression was calculated. |
| Gating strategy | Gating was determined by the Green-red fluorescence intensity to differentiate positive cells. |

☐ Tick this box to confirm that a figure exemplifying the gating strategy is provided in the Supplementary Information.

