## [Peer Review File · Nature]

Manuscript Title: Tail engagement of arrestin at the glucagon receptor

Reviewer Comments & Author Rebuttals

Reviewer Reports on the Initial Version:

Referees' comments:

Referee #1 (Remarks to the Author):

This work reports about the structural characterization via cryo-EM of the complex between a chimeric construct of the glucagon receptor with the tail of the vasopressin V2 receptor (GCGR-V2RC) and beta-arrestin 1 (barr1). Authors characterize a complex in the tail conformation, which does not show engagement of the receptor core. Although this feature has been observed in other works before, the arrangement of the complex that is described here is unprecedented. Authors discover a new arrangement of the GPCR-arr interaction. Here, the finger loop of arrestin interacts with helix VIII of the receptor and the C-edge, which had been just described as membrane anchor so far, interacts with the receptor core. Main contacts observed in the complex are supported by mutagenesis coupled to functional assays. Interesting is also that the receptor is not any more in active conformation in this arrangement. Further, the paper provides new elements supporting the notion that it is the tail conformation that guides internalization and endosomal signaling of this receptor. Overall, the findings reported here are highly relevant and of great interest to a broad audience.

The paper is very well written, the flow of information is smooth, the text in general concise. The experimental procedure to obtain the associated complex is transparently explained and the complex is well described with just the right number of details, in a clear and pleasant way accessible to a general audience. The biochemical studies on the role of tail/core conformation in GCGR internalization are convincing. References are appropriate.

I have only one concern, which is related to the choice of the authors of equipping the GCGR with the artificial V2R tail instead of keeping the wt tail. Authors do show that this construct features a similar pattern of barr recruitment and internalization as the wt receptor. However, the distribution of phosphorylation sites in V2R tail differs substantially from that of the GCGR-tail. In particular, the distal and proximal phosphorylation cluster lie much closer to each other compared to the two putative clusters in GCGR. Functional data are measured on wt receptor and are therefore reliable, but can we exclude that a different phosphorylation pattern in the C-tail alters the arrangement of the complex? I suggest at least discussing this issue.

Minor:

Page 3, line 44. Please correct the sentence about class A receptors. Class A receptors give only transient interaction with arrestin do not internalize together with arrestin.

Page 4. The Supplementary data table that is cited at this stage is not clear. What is construct 2? It should be explained either in the text or in the legend.

Page 14, lines 307-312. This hypothesis of the rearrangement of the receptor is very speculative. To my taste, stopping the sentence at line 309, before mentioning possible conformational changes at the receptor, would be more appropriate.

Referee #2 (Remarks to the Author):

Manuscript:

Chen K. et al, "Tail engagement of beta-arrestin 1 at the glucagon receptor"

Summary:

Chen et al. present the first high resolution structures of a GPCR-arrestin complex in the so-called "tail" conformation, a functionally significant binding mode of arrestins that has eluded structural characterization thus far. They present structures of the glucagon receptor bound to beta-arrestin 1, in both the apo state and bound to glucagon. In both structures, arrestin adopts a distinctive tail conformation, in which helix VIII of the receptor, as well as PIP2, form the major sites of interaction with arrestin, in contrast to previous structures of the "core" conformation, in which the intracellular pocket of the receptor plays the major role in arrestin recruitment. The authors also present functional data on an extensive set of mutants of both arrestin and the receptor supporting the interactions observed in the structure.

Main impressions:

The manuscript is well written and organized, and the observation of the tail conformation (and the first class B GPCR-arrestin complex) represents a major contribution to the field, with the caveat that it is not immediately clear to me whether the tail conformation observed here represents a family specific conformation of arrestin or whether these observations are also generalizable to class A GPCR arrestin complexes.

Specific comments:

- It is striking that such a distinct conformation is observed for arrestin in this structure, as compared to previous structures of arrestin in complex with class A receptors. The authors frame the observed conformation of arrestin as "the" tail conformation, as opposed to the previously observed "core" conformation. This implies that the observed tail conformation is generalizable – that is, that it represents the same "tail" conformation previously hypothesized for arrestin complexes of class A GPCRs such as the beta2AR. Is this the authors' viewpoint? What evidence supports this interpretation, as opposed to the interpretation that the conformation observed here is a family-specific feature of arrestin interactions with class B GPCRs? It seems to me that absent the ability to capture a class B receptor in a "core" conformation, or a class A receptor in a "tail" conformation, this point will remain somewhat ambiguous.
- The authors state that the glucagon shifts by 7Å in the extracellular direction. This implies a completely different set of interactions in this state compared to what are observed in the active state complex. Given the poor density for the glucagon and the fact that surrounding regions of the ECD are poorly ordered, can the authors be entirely sure of the register in this region? That is, is it clear that the glucagon has shifted, as opposed to the interpretation that one end of the peptide is poorly ordered?
- More specificity is needed in the image processing methods. For example, "3D classification" is referenced, but it is not clear if this refers to ab initio reconstruction, heterogeneous refinement, 3D classification without alignments, or something else, and the software used for classification is not specified. Likewise, local refinement is referred to, but the mask is not specified. The magnification used for data collection should be specified – it is not clear whether 1.071Å is the "super-resolution" pixel size or the bin2 pixel size (I suspect the latter?).
- The methods text states that 2D classes were used as templates for picking the entire dataset, but Ext. Data Fig. 1 seems to imply that projections of an ab initio reconstruction from the 500 mic subset were used. Which is the case?
- Movies of 3D variability analysis results are presented, but these could do with some more explanation – the observed changes seem rather small and difficult to interpret. How to the authors interpret these dynamics?

Author Rebuttals to Initial Comments:

Referee #1:

Remarks to the Author:

This work reports about the structural characterization via cryo-EM of the complex between a chimeric construct of the glucagon receptor with the tail of the vasopressin V2 receptor (GCGR-V2RC) and beta-arrestin 1 (barr1). Authors characterize a complex in the tail conformation, which does not show engagement of the receptor core. Although this feature has been observed in other works before, the arrangement of the complex that is described here is unprecedented. Authors discover a new arrangement of the GPCR-arr interaction. Here, the finger loop of arrestin interacts with helix VIII of the receptor and the C-edge, which had been just described as membrane anchor so far, interacts with the receptor core. Main contacts observed in the complex are supported by mutagenesis coupled to functional assays. Interesting is also that the receptor is not any more in active conformation in this arrangement. Further, the paper provides new elements supporting the notion that it is the tail conformation that guides internalization and endosomal signaling of this receptor. Overall, the findings reported here are highly relevant and of great interest to a broad audience.

The paper is very well written, the flow of information is smooth, the text in general concise. The experimental procedure to obtain the associated complex is transparently explained and the complex is well described with just the right number of details, in a clear and pleasant way accessible to a general audience. The biochemical studies on the role of tail/core conformation in GCGR internalization are convincing. References are appropriate.

— We are grateful to the reviewer for the positive assessment.

I have only one concern, which is related to the choice of the authors of equipping the GCGR with the artificial V2R tail instead of keeping the wt tail. Authors do show that this construct features a similar pattern of barr recruitment and internalization as the wt receptor. However, the distribution of phosphorylation sites in V2R tail differs substantially from that of the GCGR-tail. In particular, the distal and proximal phosphorylation cluster lie much closer to each other compared to the two putative clusters in GCGR. Functional data are measured on wt receptor and are therefore reliable, but can we exclude that a different phosphorylation pattern in the C-tail alters the arrangement of the complex? I suggest at least discussing this issue.

— We thank the reviewer for this comment. The V₂R tail was also used in the structural studies of other GPCR–arrestin complexes, including β_1 AR– β arr1 and M2R– β arr1 (Lee, Y. *et al. Nature* 583:862-866, 2020; Staus, D. P. *et al. Nature* 579:297-302, 2020). These two receptors have very different C-terminal sequences. β_1 AR has a long C terminus with multiple putative phosphorylation clusters, while the C terminus of M2R is very short with only one phosphorylation site (see figure below). However, the β arr1-bound complexes of these two receptors, where the receptor C terminus was either replaced with the V₂R tail (β_1 AR) or extended by the V₂R tail (M2R), adopt a similar core conformation. The arrestin-bound complexes of the other receptors that kept their own C termini in the structural studies, including V₂R, NTSR1, 5HT2B, and rhodopsin, also exhibit a core conformation, despite their distinct features in term of the C-terminal phosphorylation pattern (see figure below). The core conformations in all these previously determined GPCR–arrestin complexes display a similar binding mode between the receptor core and the arrestin finger loop, suggesting that the differential C-terminal tails of the receptors have little effect on the interaction pattern between the receptor core and arrestin.

In contrast to the core conformation, the GCGR^{V2RC}– β arr1 complex adopts a tail conformation with many unique features, which have been verified by our extensive functional studies on the wild-type

GCGR. In agreement with the above analysis, these data strongly imply that the V₂R tail unlikely alters the arrangement of the complex, especially the interaction patterns between the receptor helix VIII and the central loops of β arr1 as well as between the receptor helical bundle and the β arr1 C-edge. To make this clear, the statement “The tail engagement of the GCGR^{V2RC}- β arr1 complex unlikely results from the C-terminal V₂R-tail replacement of GCGR, as all the previously determined arrestin-bound GPCR structures adopt the core conformation despite different C-terminal tails in those receptors (with or without the V₂R tail)” has been added to the revised version (lines 113-116, page 5).

Helix VIII

V₂R--SSELRSLLCCARGRTPPSLGPQDESC~~TTASSSLAKDTSS~~

β ₁AR--RKA~~FQRL~~LAFPRKADRRLLHAGGQPAPLPGGFISTLGSPEHSPGGTWSDCNGGTRGGSESSLEERHSKTSRSESKMEREKNILATTRFYCTF...

M2R--KKTFKHL~~LMCHYKNIGAT~~R--V₂R tail

GCGR--WRLGKVLWEERNTSNH~~RASSSPGHGPPSKELQFGRGGGSQDSSAETPLAGGLPRLAESPF~~

NTSR1--IFLATLACLCPVWRRRRRKRPAFSRKADSVSSNH~~TLSSNATRETLY~~

5HT2B--TFRDAFGRYITCNYRATKSVKTLRKR~~SSKIYFRNPM~~AENS~~SKFFK~~KHGIRNGINPAMYQSPMRLR~~SSTIQSSSI~~ILLDTLLLTENEGDKTEEQVSYV

Rhodopsin--FRNCMLTTICCGKNPLGDDEASATVSKTETSQVAPA

Sequence replaced with V₂R tail S/T: putative phosphorylation site Sequence removed in structural study

V₂R tail sequence used in structural studies: ARGRTPPSLGPQDESC~~TTASSSLAKDTSS~~

- **The C-terminal sequences of the GPCRs with known arrestin complex structures.** The C-terminal region of helix VIII in each receptor is highlighted by an orange box. The C-terminal regions in GCGR and β ₁AR that were replaced with the V₂R tail in the structural studies are highlighted by green boxes. The V₂R tail was directly linked to the C terminus of M2R. The putative phosphorylation sites are colored red.

A binding site in the N-lobe groove of arrestin has been observed for a phosphorylation cluster in the receptor C-terminal region in some of the GPCR–arrestin structures. However, the interaction modes between the arrestin and other phosphorylation clusters and the effects of distinct phosphorylation patterns in different GPCRs on the arrangement of the receptor–arrestin complex are largely unknown. More structural and functional evidences are required to understand the molecular mechanism of the diverse phosphorylation patterns in governing arrestin recognition, which, however, is out of main scope of the current study.

Minor:

Page 3, line 44. Please correct the sentence about class A receptors. Class A receptors give only transient interaction with arrestin do not internalize together with arrestin.

— As suggested, the sentence “Based on the trafficking itineraries after internalization, the GPCRs are categorized into two classes: “class A” receptors release arrestin soon after internalization and recycle rapidly to the plasma membrane, while “class B” receptors intend to undergo sustained internalization into endosomes with the arrestin bound^{9,10}” has been changed to “Based on the trafficking itineraries after internalization, the GPCRs are categorized into two classes: “class A” receptors internalize alone after a transient interaction with the arrestin and recycle rapidly to the plasma membrane, while “class B” receptors intend to undergo sustained internalization into endosomes with the arrestin bound^{9,10}” (lines 43-47, page 3).

Page 4. The Supplementary data table that is cited at this stage is not clear. What is construct 2? It should be explained either in the text or in the legend.

— Construct 2 is the β arr1 construct that was used to determine the GCGR^{V2RC}- β arr1 structures. Both the GCGR and β arr1 constructs (construct 1 and construct 2) used for structure determination have already been defined in the legend of Extended Data Table 1 as “^{††}Construct 1, the GCGR construct that was used to determine the structures” and “^{††}Construct 2, the β arr1 construct that was used to determine the structures” (lines 819-820, page 43).

Page 14, lines 307-312. This hypothesis of the rearrangement of the receptor is very speculative. To my taste, stopping the sentence at line 309, before mentioning possible conformational changes at the receptor, would be more appropriate.

— We followed the suggestion and have removed the hypothesis of the receptor rearrangement in page 14.

Referee #2:

Remarks to the Author:

Manuscript:

Chen K. et al, “Tail engagement of beta-arrestin 1 at the glucagon receptor”

Summary:

Chen et al. present the first high resolution structures of a GPCR-arrestin complex in the so-called “tail” conformation, a functionally significant binding mode of arrestins that has eluded structural characterization thus far. They present structures of the glucagon receptor bound to beta-arrestin 1, in both the apo state and bound to glucagon. In both structures, arrestin adopts a distinctive tail conformation, in which helix VIII of the receptor, as well as PIP2, form the major sites of interaction with arrestin, in contrast to previous structures of the “core” conformation, in which the intracellular pocket of the receptor plays the major role in arrestin recruitment. The authors also present functional data on an extensive set of mutants of both arrestin and the receptor supporting the interactions observed in the structure.

Main impressions:

The manuscript is well written and organized, and the observation of the tail conformation (and the first class B GPCR-arrestin complex) represents a major contribution to the field, with the caveat that it is not immediately clear to me whether the tail conformation observed here represents a family specific conformation of arrestin or whether these observations are also generalizable to class A GPCR arrestin complexes.

— We are grateful to the reviewer for the positive assessment. Indeed, it is possible that the arrestin binds to GPCRs in family-specific manners. The class B receptors share some common structural features. They all have a long helix VIII, which is involved in coupling to the downstream signal transducers such as G proteins, which was not seen for the other GPCR families. The key role of helix VIII in the tail engagement of the GCGR^{V2RC}-βarr1 complex further highlights the importance of this region in transducer recognition for this GPCR family. Furthermore, the observed tail conformation of the GCGR^{V2RC}-βarr1 complex is largely distinct from the tail conformation previously hypothesized for the class A GPCR-arrestin complexes, suggesting that different GPCR families may adopt distinct interaction patterns with the arrestin. However, the recognition between the GPCR and arrestin is complicated with multiple elements involved. At this stage, we cannot draw a clear conclusion based on the current data. More structural information is needed to fully understand the arrestin binding behaviors of different GPCR families (please also see response to the first specific comment below).

Specific comments:

- *It is striking that such a distinct conformation is observed for arrestin in this structure, as compared to previous structures of arrestin in complex with class A receptors. The authors frame the observed conformation of arrestin as “the” tail conformation, as opposed to the previously observed “core” conformation. This implies that the observed tail conformation is generalizable – that is, that it represents the same “tail” conformation previously hypothesized for arrestin complexes of class A GPCRs such as the beta2AR. Is this the authors’ viewpoint? What evidence supports this interpretation, as opposed to the interpretation that the conformation observed here is a family-specific feature of arrestin interactions with class B GPCRs? It seems to me that absent the ability to capture a class B*

receptor in a “core” conformation, or a class A receptor in a “tail” conformation, this point will remain somewhat ambiguous.

— We thank the reviewer for this comment. As discussed in the manuscript (lines 102-106, page 5), the tail conformation of the GCGR^{V2RC}-βarr1 complex is substantially different from that previously observed in the negative-stain EM analysis of the β₂V₂R-βarr1 complex, where the tail engagement of βarr1 is solely mediated by the phosphorylated C-terminal tail of the receptor and βarr1 appears to hang from the receptor with its long axis perpendicular to the membrane plane (Shukla, A. L. *et al. Nature* 512:218-222, 2014). In stark contrast, in addition to the C-terminal tail, helix VIII of GCGR plays a major role in mediating the tail engagement of βarr1 at GCGR, and the arrestin adopts a completely different binding pose in the GCGR^{V2RC}-βarr1 complex. These differences suggest multiple tail conformations for the GPCR-arrestin complexes and highlight diversity of the arrestin binding modes in recognition of different GPCRs.

As mentioned above, GPCRs may interact with the arrestin in family-specific manners. High-resolution structures of the GPCR-arrestin complexes in a tail conformation for class A receptors and other class B receptors will certainly be helpful to draw a conclusion about whether the observed tail conformation is family-specific or generalizable to other GPCR families.

• *The authors state that the glucagon shifts by 7Å in the extracellular direction. This implies a completely different set of interactions in this state compared to what are observed in the active state complex. Given the poor density for the glucagon and the fact that surrounding regions of the ECD are poorly ordered, can the authors be entirely sure of the register in this region? That is, is it clear that the glucagon has shifted, as opposed to the interpretation that one end of the peptide is poorly ordered?*

— Indeed, the shift of glucagon in the βarr1-bound complex alters the receptor-peptide interaction pattern and impairs the stability of the GCGR-glucagon complex. This is consistent with the poorer densities for the peptide C terminus and receptor ECD in the glucagon-GCGR^{V2RC}-βarr1 complex relative to the glucagon-GCGR-G_s complex. However, by making contacts with the N-terminal region of ECD and the extracellular part of TMD, including the stalk, helix II, and the second extracellular loop (ECL2), the densities in the peptide N-terminal region are much better and allow unambiguous modelling of the peptide N terminus (Extended Data Fig. 2b; see figure below).

Furthermore, structural comparison of the glucagon-GCGR^{V2RC}-βarr1 and glucagon-GCGR-G_s complexes suggests a spatial clash between the N-terminal residue H1 of glucagon and the receptor residue R308^{5.40} in the βarr1-bound complex if the peptide were in the same binding site as that observed in the G_s-bound complex (see figure below). In the glucagon-GCGR-G_s complex, the positively charged residue H1 of glucagon binds deep to the ligand-binding pocket and repels the residue R308^{5.40} of the receptor, pushing the side chain of this basic residue away from the ligand-binding pocket, while in the glucagon-GCGR^{V2RC}-βarr1 complex, the peptide undergoes an upward movement, making space for the side chain of R308^{5.40} in the ligand-binding pocket (see figure below). Thus, the conformational differences of the residue R308^{5.40} also support the different binding modes of glucagon in the two GCGR structures. Nevertheless, we agree that the detailed distance of the peptide movement (7 Å) may not be reliable given the relatively poor densities for some of the residue side chains in the peptide N-terminal region, and thus, the distance has been removed from the revised version.

To make these clear, the statement “Comparison with the glucagon-GCGR-G_s structure reveals a shift of the peptide towards the extracellular surface in the βarr1-bound complex (Fig. 3c). The different binding modes of glucagon in the two structures are associated with distinct rotamer conformations of the receptor residue R308^{5.40} (superscript refers to the Wootten numbering system³⁸).

In the glucagon–GCGR–G_s complex, the positively charged residue H1 of glucagon binds deep to the ligand-binding pocket and repels the side chain of R308^{5.40} away from the ligand-binding pocket, while in the glucagon–GCGR^{V2RC}–βarr1 complex, the shift of the peptide makes space for the residue R308^{5.40}, allowing its side chain to point towards the center of helical bundle (Extended Data Fig. 3f) has been added to the revised version (lines 245-254, page 11). The figure showing the comparison of the glucagon binding modes and the conformations of R308^{5.40} in the two structures has been added as Extended Data Fig. 3f.

- **Comparison of the glucagon binding modes and the conformations of R308^{5.40} in the glucagon–GCGR^{V2RC}–βarr1 and glucagon–GCGR–G_s complexes.** **a**, Alignment of the two structures. Glucagon and the residue R308^{5.40} in the two structures are shown as sticks. **b**, Densities of glucagon and R308^{5.40} in the two structures. Densities are colored grey.

• *More specificity is needed in the image processing methods. For example, “3D classification” is referenced, but it is not clear if this refers to ab initio reconstruction, heterogeneous refinement, 3D classification without alignments, or something else, and the software used for classification is not specified. Likewise, local refinement is referred to, but the mask is not specified. The magnification used for data collection should be specified – it is not clear whether 1.071 Å is the “super-resolution” pixel size or the bin2 pixel size (I suspect the latter?).*

— These suggestions are well taken. We have added these details to the Methods:

“Data collection was conducted on a 300 kV Titan Krios G3 electron microscope (FEI) equipped with a Gatan K3 summit direct detection camera and a GIF-Quantum energy filter at a magnification of 81,000×. The movies were captured with a bin2 pixel size of 1.071 Å using the super-resolution counting mode of SerialEM⁵³” (lines 604-607, page 29).

“A total of 5,583 movies were collected and subjected to beam-induced motion correction using MotionCor2⁵⁴. The contrast transfer function (CTF) parameters of each micrograph were estimated using CTFFIND4 in CryoSPARC⁵⁵. The following data processing procedures were also performed by CryoSPARC⁵⁵. The particles from 500 micrographs were picked by blob picker and extracted for two rounds of two-dimensional (2D) classification. After manual selection, 190,906 particles were subjected to ab initio reconstruction and the projections of the resulting map served as a template to pick particles from the entire dataset. In total, 4,041,891 particles were picked and extracted for 2D classification. The best-looking classes of 2,531,077 particles were subjected to ab initio reconstruction for initial three-dimensional (3D) classification, generating 5 classes of initial models without any preset templates. The particles in the best-looking class were subjected to further 2D classification, ab initio reconstruction, and heterogeneous refinement. After removing the class of blurry particles, 551,645 particles were subjected to 3D classification without alignments by setting the number of classes to ten. Two sets of particles were classified, including one in ligand-bound state (300,738 particles) and the other one in ligand-free state (250,907 particles). These two datasets were subjected to non-uniform refinement and local refinement using a mask encompassing the receptor and βarr1,

resulting in two final maps with global resolutions at 3.3 Å and 3.5 Å, respectively” (lines 611-628, page 30).

The details of 3D classification have also been added to Extended Data Fig. 1a.

• *The methods text states that 2D classes were used as templates for picking the entire dataset, but Ext. Data Fig. 1 seems to imply that projections of an ab initio reconstruction from the 500 mic subset were used. Which is the case?*

— We apologize for the confusion. The projections of the ab initio reconstruction from the 500 micrographs were used as a template to pick particles from the entire dataset. The text in the Methods has been revised accordingly: “The particles from 500 micrographs were picked by blob picker and extracted for two rounds of two-dimensional (2D) classification. After manual selection, 190,906 particles were subjected to ab initio reconstruction and the projections of the resulting map served as a template to pick particles from the entire dataset” (lines 614-617, page 30).

• *Movies of 3D variability analysis results are presented, but these could do with some more explanation – the observed changes seem rather small and difficult to interpret. How to the authors interpret these dynamics?*

—The upward shift of glucagon in the glucagon–GCGR^{V2RC}–βarr1 complex breaks the receptor-peptide interaction network and thus impairs the stability of the GCGR–glucagon complex. This is consistent with the poor densities for the C terminus of glucagon and the neighboring region of the receptor ECD. The 3D variability analysis of the cryo-EM data was performed to compare the dynamics of this region in the βarr1- and G_s-bound complexes. To better reflect the difference, we modified the movies by coloring the cryo-EM maps according to chains (see figure below). In the 3D variability analysis of the glucagon–GCGR^{V2RC}–βarr1 complex, the cryo-EM data display a large variation in the C terminus of the peptide (Supplementary Video 1), suggesting a highly dynamic nature of this region in this complex. In contrast, for the G_s-bound complex, the cryo-EM data only exhibit a slight variation in the same region (Supplementary Video 2), indicating high stability of the complex.

• **Snapshots of 3D variability analysis of the cryo-EM data of glucagon–GCGR^{V2RC}–βarr1 and glucagon–GCGR–G_s.** The cryo-EM maps are colored blue (GCGR), magenta (glucagon), orange (βarr1), yellow (G_α), light pink (G_β), and pink (G_γ). The C-terminal region of glucagon and the neighboring region of the receptor ECD in the two complexes are highlighted by red boxes.

Reviewer Reports on the First Revision:

Referees' comments:

Referee #1 (Remarks to the Author):

The authors have well addressed my comments and those of the colleagues. The manuscript has improved and is not suitable for this journal. I strongly recommend its publication without delay.

Referee #2 (Remarks to the Author):

The authors have addressed my comments comprehensively. No further changes are requested.